# Ego3S: Select, Strengthen, and Synchronize for Efficient Egocentric Reasoning

**Shenshen Li**[1]  **Kaiyuan Deng**[1]  **Ruohuai Xie**[1]  **Xing Xu**[2]  **Heng Tao Shen**[2]  **Yazhou Yao**[3]  **Fumin Shen**[1]

## Abstract

Egocentric reasoning fundamentally differs from third-person understanding in LVLMs. Third-person settings offer wide and stable contexts with consistent global regularities, allowing models to utilize broad statistical correlations. In contrast, egocentric scenes are highly dynamic and heterogeneous, where decisive cues are localized and atypical. Therefore, robust egocentric reasoning requires models to focus on "*what is seen now*", *i.e.*, the immediate visual input. However, existing methods tend to exhibit "*inertial thinking*", relying excessively on language priors and global context. To address this limitation, we propose a novel three-stage **Ego3S** framework to ground models' reasoning in interaction evidence. Specifically, before training, we first utilize the counterfactual-based paradigm to **select** high-value samples that effectively activate multimodal reasoning, thus mitigating the overreliance on language priors and global context. Moreover, we introduce an interaction-centric reward for reinforcement learning that **strengthens** the model's sensitivity to localized interaction cues. Finally, during training, we employ a variance-aware learning schedule that monitors reward distributions to dynamically **synchronize** data selection with the evolving model competence. Experimental results show that our Ego3S achieves superior performance *using only 26.5% of training data, while reducing computational costs by over 46%*. Code is available at https://github.com/Mrshenshen/Ego3S.

[1]School of Computer Science and Engineering, University of Electronic Science and Technology of China, Chengdu 611731, China [2]School of Computer Science and Technology, Tongji University, Shanghai 201804, China [3]School of Computer Science and Engineering, Nanjing University of Science and Technology, Nanjing 210094, China. Correspondence to: Fumin Shen <fshen@uestc.edu.cn>.

*Proceedings of the 43rd International Conference on Machine Learning*, Seoul, South Korea. PMLR 306, 2026. Copyright 2026 by the author(s).

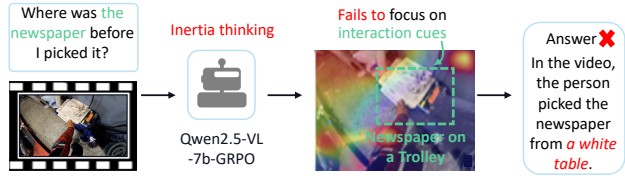

(a) An illustrative failure case of current LVLMs due to inertia thinking

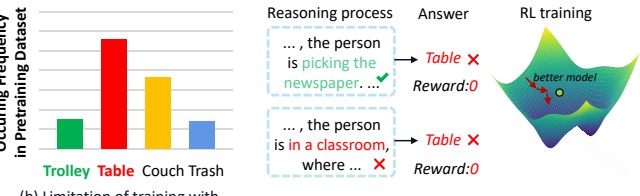

(b) Limitation of training with spurious context shortcuts

(c) Illustration of the reward–reality mismatch

*Figure 1.* Illustrative examples of existing problems: (a) A representative failure case of the recent state-of-the-art method Qwen2.5-VL-7b (Yang et al., 2024). (b) Limitation of training with spurious context shortcuts. (c) Illustration of the reward–reality mismatch in current RL objectives.

## 1. Introduction

Driven by the rapid progress in large language models (LLMs) (Touvron et al., 2023; Chiang et al., 2023; Du et al., 2022), large vision-language models (LVLMs) (Liu et al., 2023a; Bai et al., 2023; Zhu et al., 2024; Wang et al., 2024b) have recently demonstrated remarkable reasoning ability across diverse multimodal tasks. With growing interest in embodied artificial intelligence, LVLMs are being deployed in increasingly interactive and real-time domains such as robotic agents (Long et al., 2024; Liu et al., 2024; Wang et al., 2024a) and wearable vision systems (Zhang et al., 2024; Li et al., 2026; 2024b; 2025b; 2024a). In these contexts, the capacity for egocentric reasoning from a dynamic first-person perspective remains a fundamental yet underexplored challenge.

This challenge arises from the inherent gap between third-person and egocentric reasoning. In third-person settings, the visual scene is typically captured from an external observer-centric viewpoint, offering a spatially coherent and globally stable context. Under such conditions, models can effectively reason by exploiting broad statistical correlations between objects and their contextual environments. Conversely, egocentric scenarios are highly dynamic and

heterogeneous, where decisive visual cues are localized and transient rather than globally distributed. Moreover, these cues are often atypical rather than following common visual patterns, emerging mainly in human-object interaction. Essentially, *this reflects a shift from spectator cognition to actor cognition*: Third-person reasoning deciphers a world of stable states, while egocentric reasoning must navigate a world of dynamic actor-driven events.

To adapt LVLMs for egocentric scenarios, current strategies, such as adapter-tuning (Kukleva et al., 2024; Zhao et al., 2024; Li et al., 2023; Jiang et al., 2024) or direct finetuning on egocentric corpora (Chen et al., 2024a; Lin et al., 2024), often fail to prevent a crucial drawback: *inertia thinking*. Specifically, these models exhibit an over-reliance on pre-trained linguistic and contextual priors, leading them to hallucinate plausible predictions based on generic context rather than attending to the actual visual evidence, *i.e.*, "*what is seen now*". For example, as shown in Figure 1(a), the model fails to attend to interaction cues "*pick newspaper*", thus incorrectly identifying the newspaper on a "*table*" rather than a "*trolley*".

Such a drawback can be attributed to two aspects: 1) *Spurious context shortcuts*: A substantial portion of training samples can be answered correctly only using textual priors or global context. As shown in Figure 1(b), training with such samples may lead the model to develop a bias toward more common objects in its pretraining distribution, thereby failing to effectively activate egocentric reasoning based on critical interaction cues. 2) *Reward–reality mismatch*: As illustrated in Figure 1(c), current RL objectives assign identical rewards to reasoning processes of vastly different quality. Even when the model correctly attends to interaction cues "*picking the newspaper*", it receives the same reward with a prior-biased reasoning "*in a classroom*", which leads the model to overlook genuine interaction semantics.

To address these issues, we propose a novel three-stage framework termed ***Ego3S*** for enhancing egocentric reasoning by *selecting* high-value data, *strengthening* learning with vision-driven objectives, and *synchronizing* the training curriculum with the model's evolving capabilities. In the first stage, we introduce a *Counterfactual-based Selection Mechanism (CSM)* to identify high-value samples that demand genuine egocentric reasoning, thereby mitigating the model's reliance on pretraining priors. The CSM employs a two-level filtering process: 1) A coarse-grained filter discards samples solvable by language priors alone by comparing multimodal versus text-only outputs. 2) A fine-grained filter removes samples that can be solved with general visual context but without specific interaction regions by contrasting outputs from full versus interaction-masked inputs. In the second stage, we introduce *Interaction-centric Reinforcement Learning (IRL)* to strengthen attention and

reasoning over localized cues, which utilizes an explicit interaction-driven reward that forces a trade-off between statistical language priors and visual interaction semantics, thus steering the policy toward decisions grounded in the immediate egocentric observation. In the final stage, we propose a *Evolving Curriculum Recuration (ECR)* to dynamically synchronize data selection with the model's evolving competence. By tracking reward variance during training, it detects when current training samples are no longer informative and refreshes the corpora, allowing timely gains in the model's upper bound. Finally, experiments on five datasets show that Ego3S achieves state-of-the-art performance with only 3,180 samples, compared to the full dataset of 12,000 samples, while reducing training costs by over 46%.

**Why specific to egocentric reasoning?** Unlike third-person videos, where a stable and globally informative scene often serves as a reliable indicator for LVLMs (**spectator cognition**), egocentric scenarios are inherently dynamic and driven by transient human-object interactions (**actor cognition**). Therefore, in third-person settings, utilizing background priors is a valid heuristic. However, in egocentric reasoning, *such reliance amplifies "inertial thinking", leading to prior-based hallucinations rather than interaction-grounded predictions*. Our Ego3S is explicitly designed to dismantle these spurious context shortcuts via the CSM and force attention onto localized interaction cues by the IRL. While potentially beneficial for general dynamics, this mechanism is uniquely critical for egocentric scenarios (Appendix.B.5) where global context has a much lower signal-to-noise ratio and interaction cues dominate.

Overall, our contributions can be summarized as follows: (1) We propose a *Counterfactual-based Selection Mechanism (CSM)* that filters out samples solvable by language or context shortcuts, yielding a high-value dataset that explicitly requires reasoning by interaction cues. (2) We develop an *Interaction-centric Reinforcement Learning (IRL)* objective with an interaction-aware reward, bridging the gap between linguistic priors and visual evidence to strengthen egocentric reasoning. (3) We introduce an *Evolving Curriculum Recuration (ECR)* that dynamically refreshes training data based on reward variance, preventing learning stagnation and continuously raising the reasoning upper bound.

## 2. Related Work

**Large Vision-Language Models.** Large Language Models (LLMs) (Touvron et al., 2023) have evolved from text-only reasoning to multimodal understanding through the rise of LVLMs (Liu et al., 2023b; Zhu et al., 2024; Su et al., 2023). Most LVLMs follow a two-stage training paradigm, which has achieved excellent performance in various domains (Dai et al., 2023; Xu et al., 2024). However, most LVLMs mainly adopt a third-person perspective (Cheng et al., 2024), emphasizing static scene understanding. Moreover, existing

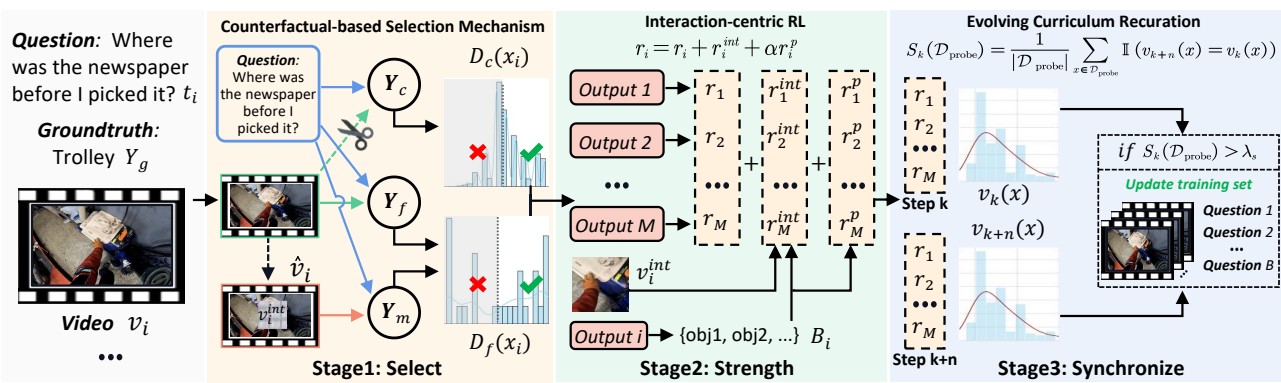

*Figure 2.* The overall framework of the proposed Ego3S method. The Ego3S first selects high-value data requiring interaction-grounded reasoning via a Counterfactual-based Selection Mechanism (CSM). It then strengthens the model's focus on visual cues using Interaction-centric Reinforcement Learning (IRL). Finally, it synchronizes the curriculum via Evolving Curriculum Recuration (ECR) to mitigate stagnation and raise the performance ceiling.

strategies, such as specific adapters (Kukleva et al., 2024; Zhao et al., 2024) or direct finetuning on egocentric corpora (Chen et al., 2024a; Lin et al., 2024), overly rely on pre-trained priors instead of focusing on interaction cues. In this paper, we focus on improving egocentric reasoning to align LVLMs more closely with human-like perception.

**Data Selection.** The conventional wisdom (Guo et al., 2025; Team et al., 2025) in training LVLMs has often emphasized data scaling, with the assumption that larger data volumes lead to better performance. However, recent methods (Li et al., 2025d;a) demonstrate that curated datasets can yield superior results in textual reasoning tasks. For example, LIMO (Ye et al., 2025) and s1 (Muennighoff et al., 2025) have shown that models trained with minimal curated samples outperform those trained on larger corpora.

**Reinforcement Learning for Reasoning.** Reinforcement learning (RL) (Yu et al., 2025; Rafailov et al., 2023; Schulman et al., 2017; Guo et al., 2025) has become a key paradigm for enhancing the reasoning abilities of models. A growing body of work is now specifically applying these techniques to enhance visual reasoning in MLLMs (Peng et al., 2025; Yang et al., 2025; Liu et al., 2025; Li et al., 2025c). However, a common thread across these methods is that the reward signal is typically a simple binary score for answer correctness. These overlook a critical issue (Wu et al., 2025): they fail to explicitly incentivize the process of visual grounding and are susceptible to learning from low-quality or shortcut-solvable samples. Therefore, we propose the Ego3S framework to integrate a principled data selection mechanism and an interaction-centric reward, ensuring that the model learns how to reason from interaction evidence.

## 3. Method

**Overview of Ego3S Method.** As shown in Figure 2, our Ego3S method aims to guide LVLMs toward reasoning grounded in interaction cues. Given a training instance

$x_i = (t_i, v_i)$, we first employ the counterfactual-based principle to quantify two-level output discrepancy between predictions under original inputs $Y_f(x)$ and counterfactual inputs, including text-only inputs $Y_c(x)$ and masked inputs $Y_m(x)$. Samples with low discrepancy are considered as prior-biased and will be discarded. Moreover, we introduce an interaction-centric reward $r_i^{int}$ that explicitly encourages the model to attend to and utilize interaction cues. Finally, to ensure sustained learning and prevent performance stagnation, we monitor reward variance $v_k(x)$ to dynamically refresh the curriculum with more challenging data, thereby enhancing the model's reasoning upper bound. Training on these high-value samples with our tailored RL objective cultivates robust egocentric reasoning.

### 3.1. Counterfactual-based Selection Mechanism

Standard training datasets inevitably contain samples that can be solved using spurious context shortcuts, where models rely on language priors or background cues. Training on these biases decisions toward statistical correlation rather than interaction evidence. Therefore, we introduce Counterfactual-based Selection Mechanism (CSM) to filter uninformative samples at two levels, obtaining a subset that requires genuine engagement with critical interaction cues.

**Coarse-grained Counterfactual-based Filter.** We first apply a coarse-grained counterfactual-based filter (CCF) to discard samples that overly rely on language priors. As shown in Figure 3(a), the model generates identical outputs in the presence or absence of visual input. Specifically, we employ the counterfactual-based paradigm to formalize the influence of visual semantics on predictions, by computing divergence under distinct treatment conditions. Given a multi-modal input $(t_i, v_i)$, we define the model's output probability under two distinct conditions, including $Y_f(t_i, v_i)$, the factual output given both text and image, and

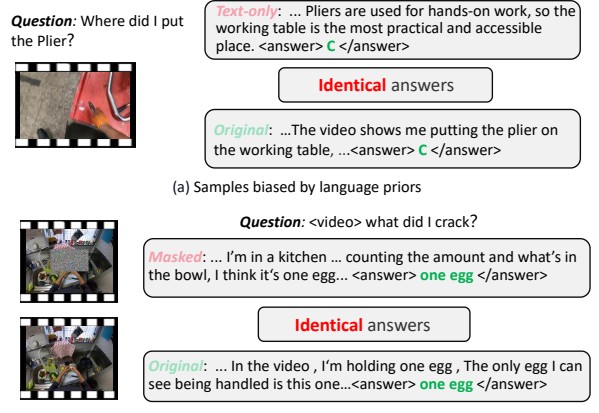

*Figure 3.* Illustrative examples for two uninformative training sample types: (a) over-reliance on language priors, and (b) excessively focusing on global context.

$Y_c(t_i)$, the counterfactual output when only the text is available. The factual output $Y_f(t_i, v_i)$ is formulated as follows:

$$y_i \sim Y_f(t_i, v_i) = \sigma\big(\log_\theta(y_i \mid t_i, v_i)\big), \quad (1)$$

The counterfactual output $Y_c(t_i)$ is formulated as follows:

$$y_i \sim Y_c(t_i) = \sigma\big(\log_\theta(y_i \mid t_i)\big), \quad (2)$$

where $\sigma(\cdot)$ denotes the softmax function.

To model the discrepancy between text-only $Y_c(t_i)$ and original multi-modal outputs $Y_f(t_i, v_i)$, we measure each condition's agreement with the ground truth $Y_g$, where the value is set to 1 when the prediction matches the ground truth, and to 0 otherwise. The coarse-grained discrepancy $D_c(x)$ for each sample $x_i$ is then defined as:

$$D_c(x_i) = \frac{1}{M} \sum_{i=1}^{M} \left[ \mathbb{I}(Y_f(t_i, v_i) = Y_g^i) - \mathbb{I}(Y_c(t_i) = Y_g^i) \right], \quad (3)$$

where $M$ is the number of rollout outputs in GRPO, and $\mathbb{I}(\cdot)$ is the indicator function (1 if the predicate holds, 0 otherwise). Finally, a sample is classified as overly relying on language priors and is then filtered if $D_c(x_i) < \mu_c + \lambda_c \sigma_c$, where $\mu_c$ and $\sigma_c$ are the corpus-level mean and standard deviation of $D_c(x)$, and $\lambda_c$ is a tunable hyperparameter.

**Fine-Grained Counterfactual-based Filter.** While the CCF eliminates language-based shortcuts, a more subtle visual shortcut remains: models can exploit general scene context rather than decisive interaction cues. For instance, as shown in Figure 3(b), a model might correctly answer a query about "*one egg*" by simply detecting its presence in a kitchen, while failing to ground its reasoning in the specific egg being held. To address this, we introduce a fine-grained counterfactual-based filter (FCF) to remove samples solvable from the global context rather than interaction regions. Specifically, we define factual and counterfactual outputs,

where the former follows Eq.1. For the counterfactual, we generate candidate local regions using sliding windows, select the one with the highest CLIP similarity to the query as the interaction region $v_i^{int}$, and obtain the counterfactual output $Y_m(t_i, \hat{v}_i)$ by masking this region in the image $\hat{v}_i$:

$$y_i \in Y_m(t_i, \hat{v}_i) \sim \text{softmax}\left[\log_\theta(y_i \mid t_i, \hat{v}_i)\right], \quad (4)$$

Following Eq.2, the output discrepancy between original $Y_f(t_i, v_i)$ and masked visual inputs $Y_m(t_i, \hat{v}_i)$ is defined as:

$$D_f(x_i) = \frac{1}{M} \sum_{i=1}^{M} \left[ \mathbb{I}(Y_f(t_i, v_i) = Y_g^i) - \mathbb{I}(Y_m(t_i, \hat{v}_i) = Y_g^i) \right], \quad (5)$$

a sample is considered not using interaction cues and then removed if $D_f(x) < \mu_f + \lambda_f \sigma_f$, where $\mu_f$ and $\sigma_f$ are the corpus-level mean and standard deviation of $D_f(x_i)$.

### 3.2. Interaction-centric RL

Despite recent progress, existing RL methods (Guo et al., 2025; Yu et al., 2025; Ahmadian et al., 2024) fail to explicitly incentivize the visual interaction. This creates a reward gap between textual and visual signals, especially in dynamic egocentric scenarios. After enhancing egocentric reasoning through data selection, we further strengthen it with an explicit interaction-driven reward in RL.

**Background of GRPO.** Building upon the GRPO (Guo et al., 2025), we retain the GRPO objective and add an interaction-centric reward that incentivizes grounding in the interaction region. For a given query, the model generates a group of $M$ outputs $\{o_1, \ldots, o_M\}$, and then optimizes the policy by maximizing the following objective:

$$\mathcal{J}_{\text{GRPO}}(\theta) = \mathbb{E}[q \sim P(Q), \{o_i\}_{i=1}^{M} \sim \pi_{\theta_{\text{old}}}(O|q)]$$

$$\frac{1}{M} \sum_{i=1}^{M} \frac{1}{|o_i|} \sum_{t=1}^{|o_i|} \left\{ \frac{\pi_\theta(o_{i,t}|q, o_{i,<t})}{\pi_{\theta_{\text{old}}}(o_{i,t}|q, o_{i,<t})} \hat{A}_{i,t} - \beta D_{\text{KL}}[\pi_\theta || \pi_{\text{ref}}] \right\},$$

where the reward $r_i$ for each output $o_i$ is a binary score based on the final answer's correctness. The advantage $\hat{A}_{i,t}$ is then calculated by normalizing this score relative to the other responses in the group $\mathbf{r} = \{r_1, \ldots, r_M\}$:

$$\hat{A}_{i,t} = \tilde{r}_i = \frac{r_i - \text{mean}(\mathbf{r})}{\text{std}(\mathbf{r})}. \quad (6)$$

**Interaction-centric Rewards.** To bias the policy toward interaction-grounded reasoning rather than pretrained priors, we reward the model for textually mentioning objects that are verifiably present within the interaction region. First, during each rollout, we extract all objects $\mathcal{B}_i$ from the model's response and compute their CLIP similarity $S_o$ with the ground-truth interaction region $v_i^{int}$ obtained by the FCF. Based on these, the initial interaction-centric reward $r_i^{int}$ can be formulated as:

$$r_i^{int} = \max(H - \frac{1}{M} \sum_{i=1}^{M} \mathbb{I}(S_o^i > \lambda_o), 0) \cdot \mathbb{I}\left(\exists i : S_o^i > \lambda_o\right), \quad (7)$$

This term acts as a targeted incentive, rewarding the act of engaging with interaction cues to create a better trade-off between reasoning from immediate visual semantics and pre-trained priors. For instance, consider an image of a person taking a bottle out of a refrigerator, with the query, "*What drink is the person getting?*" A model governed by priors will likely answer "*milk*" or "*juice*", relying on the strong statistical correlation between "*refrigerator*" and common "*beverages*". It makes a plausible guess without ever needing to see the specific details of the interacting bottle. This reward can fundamentally alter this behavior, which provides a positive reward for attention to the interaction bottle's label. It lowers the activation energy for pretraining priors. When the model is incentivized to focus on the interaction, it may discover the label reads "*Soy Sauce*", enabling it to overcome prior reliance and focus on interaction evidence.

However, to prevent the policy from generating superfluous text or inefficiently using visual operations, we introduce a penalty term $r_i^p$, which can be formulated as follows:

$$r_i^p = \min(N - n_{obj}(\mathcal{B}_i), 0), \tag{8}$$

where $n_{obj}(\mathcal{B}_i)$ is the number of objects in $\mathcal{B}_i$. Such a $r_{\text{penalty}}(y)$ acts as an efficiency penalty, discouraging redundant object extraction by comparing the number of extracted objects, $n_{obj}(\mathcal{B}_i)$, relative to a desired maximum $N$.

Finally, we incorporate the interaction-centric reward $r_i^{int}$ and the penalty term $r_i^p$ into the standard GRPO reward, forming the objective of IRL. The final reward is defined as:

$$r_i = r_i + r_i^{int} + \alpha r_i^p, \tag{9}$$

where $\alpha$ is a scaling coefficient controlling the influence of the penalty term.

### 3.3. Evolving Curriculum Recuration

While the data refined by our CSM is effective for initiating egocentric reasoning, there still remains a fundamental challenge in subsequent training: as the model's competence improves, a portion of the data becomes "*mastered*" and thus uninformative, leading to performance stagnation. To ensure the model can continuously push its performance ceiling, we introduce the Evolving Curriculum Recuration (ECR) to dynamically synchronize the training curriculum with the model's evolving ability.

Inspired by recent work (Li et al., 2025d), we find that a curriculum becomes uninformative when a significant portion of its samples consistently yield the same rewards across training stages. Therefore, we use reward variance dynamics to determine when the entire training curriculum needs to be refreshed. Specifically, we first designate a fixed probing set $\mathcal{D}_{\text{probe}}$, composed of training data from the initial $n$ steps. This set acts as a consistent benchmark. At periodic monitoring intervals, specifically step $k$ and step $k + n$, we evaluate the current policy on this same probing set. For each sample $x \in \mathcal{D}_{\text{probe}}$, the RL yields a set of rewards $\{r_i\}_{i=1}^M$ from $M$ rollouts, from which we compute the sample-wise reward variance of the k-th training step:

$$v_k(x) = \text{Var}\big(\{r_i\}_{i=1}^M\big), \tag{10}$$

Then, we define the stagnation ratio $S_k$, as the fraction of samples in the probing set whose reward variance has remained unchanged between step $k$ and step $k + n$. This metric directly quantifies the portion of the curriculum from which the model has ceased to learn:

$$S_k(\mathcal{D}_{\text{probe}}) = \frac{1}{|\mathcal{D}_{\text{probe}}|} \sum_{x \in \mathcal{D}_{\text{probe}}} \mathbb{I}\big(v_{k+n}(x) = v_k(x)\big). \tag{11}$$

If this ratio $S_k(\mathcal{D}_{\text{probe}})$ exceeds a predefined stagnation threshold $\lambda_s$, the ECR module triggers a full recuration of the training data. When triggered, the current training set is discarded, and a more challenging curriculum is generated by re-applying our entire data selection pipeline (starting from CSM) on a larger data pool using the latest model parameters. This feedback loop ensures computational resources are consistently focused on data at the frontier of the model's capabilities and prevents training stagnation.

## 4. Experiments

### 4.1. Experimental Setup

**Training Datasets.** Main results in Tables are based on models trained with the data constructed from the Ego4D dataset (Grauman et al., 2022), a high-quality multi-modal egocentric dataset. Specifically, we construct a subset of 12,000 samples from the Ego4D, covering both open-ended QA (OpenQA) and close-ended QA (CloseQA) tasks.

**Evaluation.** Similar to (Tian et al., 2025), we evaluate models on both OpenQA and CloseQA tasks using the $pass@1$ metric, where $pass@1$ denotes the percentage of problems correctly solved on the first attempt in a zero-shot setting. For the OpenQA task, we assess the model's ability on three benchmarks: QAEgo4D (Bärmann & Waibel, 2022), EgoTextVQA (Zhou et al., 2025), and EgoSchema (Mangalam et al., 2023). In terms of the CloseQA task, we evaluate on Egoplan2 (Qiu et al., 2024) and Egothink (Cheng et al., 2024) benchmark datasets.

**Implementation Details.** Following prior methods (Feng et al., 2025; Yao et al., 2024), we conduct our main experiments on egocentric reasoning tasks, using Qwen2.5-VL-7B (Yang et al., 2024) as the base model. First, we apply CSM to select high-value samples using the initial model before training. Moreover, we introduce the IRL to guide the model toward focusing on interaction cues. The selected samples are then used for model training with the AdamW optimizer

| Method | Time ↓ | Egoplan2 | QAEgo4D | EgoTextVQA | EgoSchema | Egothink |
|---|---|---|---|---|---|---|
| Qwen2.5-VL-7b-Full | 60.5 | 30.43 | 63.80 | 20.30 | 47.60 | 56.57 |
| Qwen2.5-VL-7b-s1 (2025) | 52.3 | 29.97 | 64.40 | 19.80 | 47.90 | 56.32 |
| Qwen2.5-VL-7b-LIMO (2025) | 53.7 | 30.59 | 64.20 | 20.20 | 47.40 | 56.27 |
| Qwen2.5-VL-7b-LIMR (2025) | 103.2 | 31.46 | 65.20 | 20.60 | 48.20 | 56.93 |
| Qwen2.5-VL-7b-CSM (Ours) | 29.7 | 33.76 | 65.80 | 21.40 | 48.60 | 57.23 |
| Qwen2.5-VL-7b-GRPO (2025) | 60.5 | 30.43 | 63.80 | 20.30 | 47.60 | 56.57 |
| Qwen2.5-VL-7b-RLOO (2025) | 70.8 | 30.19 | 63.20 | 19.40 | 46.90 | 55.49 |
| Qwen2.5-VL-7b-DAPO (2025) | 62.4 | 30.87 | 64.60 | 20.60 | 47.40 | 57.24 |
| Qwen2.5-VL-7b-IRL (Ours) | 62.1 | 35.42 | 66.40 | 21.80 | 48.80 | 58.22 |
| **Qwen2.5-VL-7b-Ego3S (Ours)** | **32.6** | **36.83** | **66.80** | **22.40** | **49.20** | **58.71** |

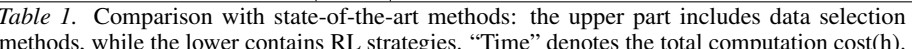

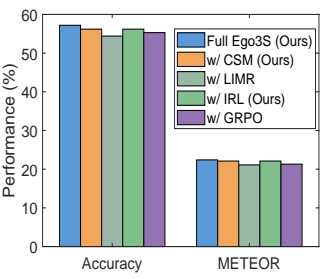

*Table 1.* Comparison with state-of-the-art methods: the upper part includes data selection methods, while the lower contains RL strategies. "Time" denotes the total computation cost(h).

*Figure 4.* Comparison with recent methods using InternVL3-2b.

and a learning rate of 1e-6. All experiments are performed for one epoch with a batch size of 512 across 8 A6000s. Finally, during training, we adopt ECR to dynamically synchronize the curriculum with the evolving ability of model.

### 4.2. Overall Comparison Results

**Comparing Methods.** We compare our approach with 1) Data selection methods, including s1 (Muennighoff et al., 2025) (LLM-based quality scoring), LIMO (Ye et al., 2025) (difficulty-aware sampling), and LIMR (Li et al., 2025d) (learning impact measurement). Moreover, we also evaluate models trained on the full dataset (Full) as the baseline. 2) RL strategies, including GRPO (Guo et al., 2025), DAPO (Yu et al., 2025) that enhances GRPO with post-rollout filtering, and RLOO (Ahmadian et al., 2024) that combines online and offline data.

**Comparisons with Data Selection Methods.** We compare Ego3S against recent data selection methods, including LIMO (Ye et al., 2025), s1 (Muennighoff et al., 2025), and LIMR (Li et al., 2025d). The results in upper part of Table 1 highlight two key findings: 1) The Ego3S consistently outperforms the baseline of training on the full dataset across all benchmarks. Remarkably, this superior performance is achieved while using only 26.5% of the data, reducing training time by over 46% and supporting our findings that not all samples are effective for activating egocentric reasoning. 2) The Ego3S achieves a 2.30% improvement over the latest method LIMR (Li et al., 2025d) on the Egoplan2 (Qiu et al., 2024). These results show that, while LIMR leverages learning dynamics for sample selection, our CSM provides a more effective strategy for egocentric scenarios by explicitly prioritizing samples that require interaction-based reasoning, thereby better capturing complex interactions. Further details on the number of samples selected by different methods are provided in the Appendix.B.3.

**Comparisons with Recent RL Strategies.** To validate the efficacy of our proposed Interaction-centric RL, we compare the Ego3S against several leading RL strategies. As shown in lower part of Table 1, we list the following conclusions: 1) Even without the proposed CSM, the IRL consistently

| No. | CSM | | IRL | | ECR | QAEgo4D | EgoTextVQA | EgoSchema |
|---|---|---|---|---|---|---|---|---|
| | CCF | FCF | $r^{int}$ | $r^p$ | | | | |
| 0 | - | - | - | - | - | 63.80 | 20.30 | 47.60 |
| 1 | ✔ | - | - | - | - | 64.80 | 20.80 | 48.20 |
| 2 | - | ✔ | - | - | - | 64.60 | 20.60 | 47.80 |
| 3 | ✔ | ✔ | - | - | - | 65.80 | 21.40 | 48.60 |
| 4 | ✔ | ✔ | ✔ | - | - | 66.20 | 21.60 | 48.60 |
| 5 | ✔ | ✔ | ✔ | ✔ | - | 66.20 | 22.00 | 48.80 |
| 6 | ✔ | ✔ | ✔ | ✔ | ✔ | **66.80** | **22.40** | **49.20** |

*Table 2.* Ablation study of Ego3S using Qwen2.5-VL-7B.

outperforms the standard GRPO (Guo et al., 2025) across all benchmarks. This improvement indicates that explicit rewarding interaction grounding is essential for guiding the policy toward stable egocentric reasoning. 2) The proposed IRL also surpasses the state-of-the-art RL method DAPO (Yu et al., 2025), which struggles to balance pretrained language priors and interaction-specific semantics.

**Effectiveness of Ego3S on Different Base Models.** As shown in Figure 4, our method consistently surpasses other recent data selection and RL methods when applied to the base model InternVL3-2b (Chen et al., 2024b). This improvement shows that Ego3S is generalized well beyond a specific architecture. The key reason lies in its design: the CSM selects samples that inherently require egocentric reasoning, while the IRL encourages the policy to ground decisions on interaction cues rather than architecture-dependent heuristics. Together, these components enable Ego3S to enhance reasoning performance across diverse LVLMs without relying on model-specific inductive biases.

### 4.3. Further Analysis

**Ablation Study.** As presented in Table 2, we list the following conclusions: 1) The results of No.0 show that training on the full dataset (w/o CSM) leads to a clear performance drop, confirming its role in promoting interaction-grounded reasoning. Comparisons between No.0/No.1 and No.0/No.2 further show that CFM and FFM effectively reduce overreliance on language priors and global context, respectively. 2) Training with a standard GRPO objective (w/o IRL) results in a substantial performance decline, even on our curated data. This proves that our interaction-centric reward

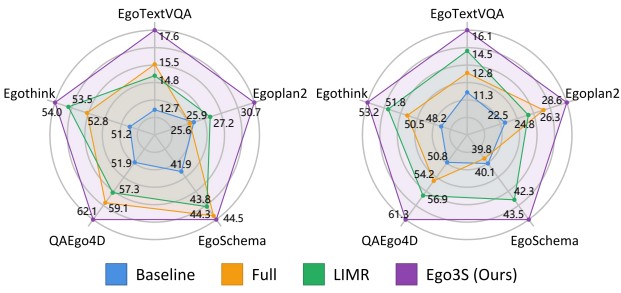

*Figure 5.* Cross-model generalization of high-value samples selected by CSM. Performance with Qwen2.5-VL-7b trained on samples from InternVL3-2b (left), and vice versa (right).

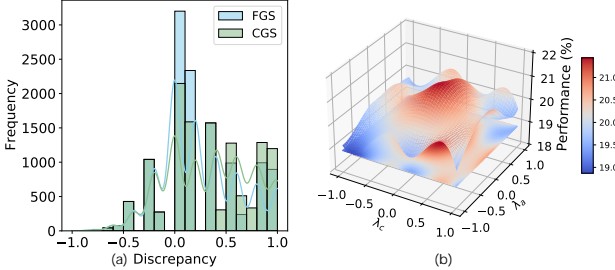

*Figure 6.* (a) Visualization of output discrepancies on the full training dataset, quantified using the proposed CCF and FCF. (b) Performance variation with respect to the hyperparameters $\lambda_c$ and $\lambda_f$ of CSM on EgoTextVQA (Zhou et al., 2025).

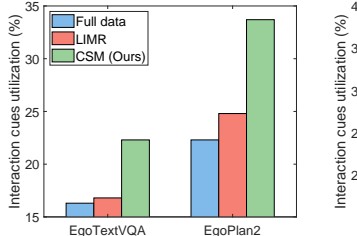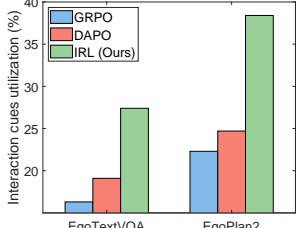

*Figure 7.* Comparative analysis of interaction cue utilization across different data selection and RL methods.

is essential for guiding the model toward visual grounding. Ablating the Penalty Reward $r^p$ also degrades performance, confirming its role in preventing inefficient visual operations. 3) Finally, a model trained on a static dataset (w/o ECR) performs worse than our full model. This validates the effectiveness of the ECR module in preventing learning stagnation by dynamically updating the curriculum to continuously challenge the model.

**Comparison on Cross-model Generalization.** To examine whether the samples identified by our CSM are model-specific or exhibit cross-architecture generality, we conduct a cross-model transfer experiment. Specifically, data curated by the Qwen2.5-VL-7B model (Yang et al., 2024) using CSM is used to train the architecturally distinct InternVL-2B model (Chen et al., 2024b), and vice versa. As illustrated in Figure 5, our Ego3S framework consistently outperforms the state-of-the-art LIMR (Li et al., 2025d) across both transfer directions, demonstrating that the high-value samples selected by CSM retain their effectiveness even when applied to different model architectures. This finding reveals two key insights. First, the high-value samples captured by CSM are not bound to the specific feature space of a single LVLM, but rather encode generalizable reasoning patterns that stimulate cross-modal alignment and interaction grounding across models. Second, the robustness of transfer performance suggests that CSM isolates samples reflecting intrinsic cognitive challenges, such as causal dependency, visual disambiguation, and intent inference—rather than model-dependent idiosyncrasies. These results highlight the universality and transferability of our selection mechanism, confirming that Ego3S not only enhances within-model reasoning but also provides a scalable foundation for general egocentric understanding across LVLM architectures.

**Comparison on Interaction Cues Utilization.** To further demonstrate the effectiveness of Ego3S, we evaluate the model's ability to leverage essential interaction cues. The interaction cue utilization is defined as the percentage of test samples that the model answers correctly with the full visual input, but fails when the key interaction region is masked.

As shown in Figure 6(a), we visualize the discrepancy distributions for both CCF (multimodal vs. text-only) and FCF (full vs. masked image). The low-scoring tails of these distributions, representing samples where language priors or general visual context are sufficient to solve the problem, are removed by our proposed CSM. Additionally, results in Figure 6(b) suggest that optimal performance is achieved with $\lambda_c = 0.1$ and $\lambda_f = 0.5$. Furthermore, as illustrated in Figure 7, our Ego3S method achieves higher interaction cue utilization compared to both GRPO (Guo et al., 2025) and the latest selection method LIMR (Li et al., 2025d). This finding suggests that Ego3S mitigates over-reliance on spurious shortcuts by filtering out uninformative samples and introducing a specific interaction-centric reward.

**Qualitative Analysis.** We conduct a qualitative analysis by comparing the reasoning processes of our Ego3S with the latest LIMR method (Li et al., 2025d). As shown in Figure 8(a), the LIMR exhibits failures in reasoning, overlooking key interaction information "*picking up*" and propagating these errors through subsequent inference steps. Conversely, our Ego3S method correctly identifies the relevant object and action, thus integrating them to arrive at the correct solution. This comparison underscores a critical distinction: while the LIMR focuses on removing easy samples, our CSM explicitly curates samples that train the reasoning skill from interaction cues. Furthermore, as presented in Figure 8(b), the IRL provides explicit supervision on interaction cues, leading to more robust behavior than the GRPO.

*Question:* What am I doing now?
*Groundtruth:* plug a power cord

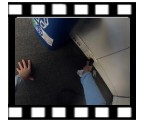

*LIMR*: <think>...My left hand is resting on the floor, and my right hand is holding a black object, possibly a phone or a remote. **I seem to be in the process of picking up or placing the object.** </think> <answer> **I am picking up a black object from the floor.** </answer>

*Ego3S (Ours)*: <think> .... I see my right hand holding what looks like a plug or small tool near the side of a desk ... I'm interacting with an outlet or connector , **seem to insert or remove a power cord or plug from the side of the furniture.** </think> <answer> **I am plugging a power cord.** </answer>

(a)

*Question:* Where were paint bottles in the artist room?
*Groundtruth:* On shelves and tables.

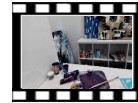

*GRPO*: <think> ... The video shows an artist room there is a white shelving unit with multiple compartments . **The paint bottles are placed in one of the compartments of this shelving unit.**</think><answer> **On shelves** </answer>

*Ego3S (OURS)*: <think> ... **on the table, there are tubes of paint, a palette, and some containers** ... Let me see if there are any other places where paint bottles exist. **At the white shelf, there are several paint bottles.**</think><answer> **On the table and on the white shelf** </answer>

(b)

*Figure 8.* Comparison of reasoning processes: (a) Ego3S versus the selection method LIMR (Li et al., 2025d); (b) Ego3S versus the GRPO (Guo et al., 2025).

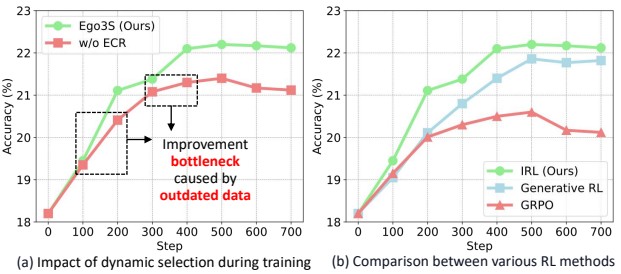

*Figure 9.* (a) Effect of the proposed ECR on upper bound. (b) Exploration of potential RL methods with generative rewards.

### 4.4. Key Insights and Discussion

**Effect of ECR on Reasoning Upper Bound.** To assess the effectiveness of our dynamic curriculum strategy, we compare the full Ego3S model with a variant trained on a static dataset, as shown in Figure 9(a). The model without ECR initially achieves rapid gains but soon encounters a clear improvement bottleneck around the 400-step mark, where performance plateaus despite continued training. This stagnation indicates that a fixed dataset gradually becomes outdated as the model's competence improves, offering limited new information for further learning and thus constraining the reasoning upper bound. In contrast, the full Ego3S model, equipped with ECR, continuously monitors learning progress and dynamically refreshes the curriculum with more challenging samples. This adaptive re-selection effectively mitigates stagnation, sustains steady performance gains, and enables the model to reach a higher final accuracy. These results confirm that ECR plays a pivotal role in maintaining data relevance throughout training and in driving continual improvement in egocentric reasoning ability.

**Why Does Less Data Outperform the Full Dataset?** Our

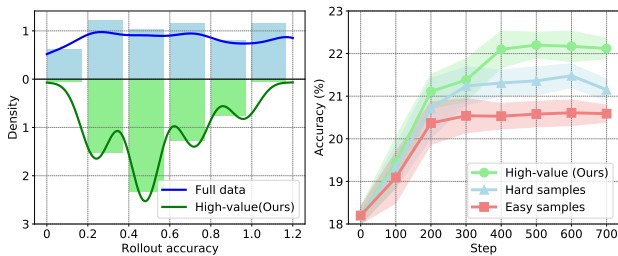

*Figure 10.* (a) Proportion of full and high-value samples across different reasoning difficulty levels. (b) Comparison of reasoning performance among high-value, hard, and easy samples.

empirical results reveal a compelling phenomenon: training with a small curated subset of data can outperform the full dataset. To further elucidate the underlying mechanisms behind this "*less is more*" phenomenon, we first analyze the distributional differences between the full data and CSM-selected high-value subset by measuring the model's average success rate across multiple rollouts. As shown in Figure 10(a), high-value samples contain far fewer easy instances than the full dataset. Consequently, in conventional large-scale training, typically limited to only one or two epochs, the model receives insufficient repeated exposure to these challenging samples, constraining the potential gains from such valuable instances. However, is difficulty the only factor? To investigate this, we train separate models on datasets consisting solely of easy, hard, and high-value samples. The results in Figure 10(b) show that while training on hard samples is better than on easy ones, it still falls short of the high-value data selected by the CSM. This suggests that our CSM is not a simple difficulty filter, which aims to isolate samples requiring genuine interaction-grounded reasoning rather than over-reliance on pretraining priors.

**Discussion on Potential Optimization for RL.** While our IRL objective improves performance by introducing an explicit reward for visual grounding, it still suffers from an inherent limitation: the reward is a heuristic binary signal. According to Goodhart's Law (Strathern, 1997), when a measure becomes a target, it ceases to be a reliable measure. This vulnerability, referred to as *"reward hacking"* in RL (Wu et al., 2025; Huang et al., 2025), is particularly pronounced in egocentric settings, where capturing the dynamic and highly variable behavior necessary for robust generalization is essential. For instance, it may assign the same reward to a correct interaction regardless of whether the model's subsequent reasoning is coherent or meaningless. This exposes a weakness in which the policy exploits flaws in the reward signal, achieving higher scores without genuinely improving its reasoning quality or generalization ability. Building on recent advances in reward modeling (Wu et al., 2025), as shown in Figure 9(b), we explore whether the generative reward can be more suitable for RL in egocentric scenarios. In this setting, the reward is defined as the proba-

bility that a LVLM outputs a positive response (*e.g.*, "*yes*") to a prompt such as: "*Does the model's explanation correctly use the visual evidence from the interaction to support its conclusion?*" This reward can incorporate uncertainty estimates from the LVLM, allowing it to weigh actions based on both correctness and confidence. Therefore, such a generative paradigm may effectively reduce the likelihood of reward hacking, leading to better generalization.

## 5. Conclusion

In this work, we introduced the three-stage Ego3S framework to enhance the egocentric reasoning in LVLMs. By incorporating the Counterfactual Selection Mechanism (CSM), we first filtered out shortcut-solvable samples for activating genuine egocentric reasoning. Moreover, we designed an Interaction-centric RL (IRL) to guide the policy toward grounding its decisions in specific interaction cues. Finally, we proposed the Evolving Curriculum Recuration (ECR) to prevent learning stagnation. For future work, we will explore the integration of generative reward models to provide a more holistic and nuanced learning signal.

## Acknowledgment

This work was supported by the National Natural Science Foundation of China under Grant (No. 62476201), the Sichuan Top Youth Scientific and Technological Talent Project under Grant (No. DQ202415), the Central Guidance on Local Science and Technology Development Fund of Shanghai City (No. YDZX20253100002004), the New Cornerstone Science Foundation through the XPLORER PRIZE, and the Fundamental Research Funds for the Central Universities.

## Impact Statement

This paper presents work whose goal is to advance the field of Machine Learning, specifically in enhancing the reasoning capabilities of embodied AI and wearable assistants. By filtering uninformative samples and optimizing data efficiency, our Ego3S framework significantly reduces computational costs and energy consumption, aligning with the goals of Green AI. While improved egocentric understanding holds promise for positive applications such as elderly care and augmented reality aids, it also raises potential privacy concerns regarding the automated analysis of first-person video in private environments. We encourage researchers and practitioners to prioritize privacy protection and transparency when deploying such systems.

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

# A. Theoretical Justification of Ego3S

## A.1. Preliminaries and Consistent Notation

For a training instance $x_i = (t_i, v_i)$, the model induces (via softmax) a predictive distribution over answers:

$$\mathbf{Y}_f(x_i) := \sigma\big( \log_\theta(y \mid t_i, v_i) \big), \tag{12}$$

$$\mathbf{Y}_c(x_i) := \sigma\big( \log_\theta(y \mid t_i) \big), \tag{13}$$

$$\mathbf{Y}_m(x_i) := \sigma\big( \log_\theta(y \mid t_i, \hat{v}_i) \big), \tag{14}$$

where $\hat{v}_i$ masks the interaction region (as in Sec. 3.1), and $\sigma(\cdot)$ is softmax. Let $Y_g^i$ denote the ground-truth answer for $x_i$. The discrepancies (averaged over $M$ rollouts) are

$$D_c(x_i) = \frac{1}{M} \sum_{j=1}^M \Big[ \mathbb{I}(\hat{y}_f^{(j)} = Y_g^i) - \mathbb{I}(\hat{y}_c^{(j)} = Y_g^i) \Big], \tag{15}$$

$$D_f(x_i) = \frac{1}{M} \sum_{j=1}^M \Big[ \mathbb{I}(\hat{y}_f^{(j)} = Y_g^i) - \mathbb{I}(\hat{y}_m^{(j)} = Y_g^i) \Big], \tag{16}$$

where $\hat{y}_f^{(j)}, \hat{y}_c^{(j)}, \hat{y}_m^{(j)}$ are sampled/decoded outputs under $\mathbf{Y}_f, \mathbf{Y}_c, \mathbf{Y}_m$ in the $j$-th rollout.

## A.2. Information-Theoretic Justification for CSM via Conditional Mutual Information

**Classic identity (conditional mutual information as expected KL).** Let $V$ be the visual input and $T$ the text input. A standard information-theoretic identity states

$$I(Y; V \mid T) = \mathbb{E}_{t,v}\Big[ \mathrm{KL}\big(p(\cdot \mid t, v) \,\|\, p(\cdot \mid t)\big) \Big]. \tag{17}$$

Likewise, if we conceptually decompose $v = (v^{\mathrm{int}}, v^{\mathrm{ctx}})$ into interaction evidence and global context, then

$$I(Y; V^{\mathrm{int}} \mid T, V^{\mathrm{ctx}}) = \mathbb{E}\Big[ \mathrm{KL}\big(p(\cdot \mid t, v^{\mathrm{int}}, v^{\mathrm{ctx}}) \,\|\, p(\cdot \mid t, \varnothing, v^{\mathrm{ctx}})\big) \Big]. \tag{18}$$

Eq. (17)–(18) are classical and hold without additional assumptions.

**Model-based instantiation.** Assuming the model posterior $\mathbf{Y}_\bullet$ approximates the true posterior on the training support, we obtain the practical proxies

$$I(Y; V \mid T) \approx \mathbb{E}\Big[ \mathrm{KL}\big(\mathbf{Y}_f(x) \,\|\, \mathbf{Y}_c(x)\big) \Big], \tag{19}$$

$$I(Y; V^{\mathrm{int}} \mid T, V^{\mathrm{ctx}}) \approx \mathbb{E}\Big[ \mathrm{KL}\big(\mathbf{Y}_f(x) \,\|\, \mathbf{Y}_m(x)\big) \Big]. \tag{20}$$

Therefore, CSM is principled: it aims to retain samples where removing vision (coarse) or removing interaction evidence (fine) significantly changes the predictive distribution, i.e., where the corresponding conditional mutual information is large.

**Why the indicator-based $D_c, D_f$ are legitimate proxies.** The filters use accuracy gaps rather than KL gaps. This is supported by a standard inequality chain connecting distributional shift to changes in event probabilities.

**Lemma 1 (Pinsker; classical).** For any distributions $P, Q$, the total variation satisfies

$$\|P - Q\|_{\mathrm{TV}} \leq \sqrt{\tfrac{1}{2}\mathrm{KL}(P\|Q)}. \tag{21}$$

**Lemma 2 (Accuracy gap lower-bounds TV).** Let $P, Q$ be two predictive distributions over answers, and consider the "correctness event" $A = \{Y = Y_g\}$. Then

$$\big|P(A) - Q(A)\big| \leq \|P - Q\|_{\mathrm{TV}}. \tag{22}$$

*Proof.* By definition, $\|P - Q\|_{\text{TV}} = \sup_B |P(B) - Q(B)|$, hence it upper-bounds the difference on any event $A$. □

Combining Lemma (22) with Pinsker (21) yields

$$\left| P(A) - Q(A) \right| \leq \sqrt{\tfrac{1}{2}\text{KL}(P\|Q)}. \tag{23}$$

Thus, a non-trivial correctness gap between $\mathbf{Y}_f$ and $\mathbf{Y}_c$ (coarse) or between $\mathbf{Y}_f$ and $\mathbf{Y}_m$ (fine) implies a non-trivial KL gap, which by Eq. (17)–(18) implies non-trivial conditional mutual information. This provides a classical information-theoretic foundation for using $D_c$ and $D_f$ as practical selection signals.

### A.3. Causal Justification for CSM as Counterfactual Necessity

**Potential-outcome view (classic causal inference).** Interpret "masking" as an intervention that removes interaction evidence. Define a binary treatment variable $A \in \{0, 1\}$: $A = 1$ corresponds to the factual input (full vision), and $A = 0$ corresponds to the counterfactual input (text-only for coarse, or interaction-masked for fine). Let $C_\bullet$ denote the correctness indicator under a specific condition:

$$C_f(x_i) := \mathbb{I}(\hat{y}_f = Y_g^i), \quad C_c(x_i) := \mathbb{I}(\hat{y}_c = Y_g^i), \quad C_m(x_i) := \mathbb{I}(\hat{y}_m = Y_g^i). \tag{24}$$

**Definition (individual treatment effect on correctness).** For coarse and fine counterfactuals respectively,

$$\text{ITE}_c(x_i) := \mathbb{E}\big[C_f(x_i) - C_c(x_i) \mid x_i\big], \tag{25}$$

$$\text{ITE}_f(x_i) := \mathbb{E}\big[C_f(x_i) - C_m(x_i) \mid x_i\big]. \tag{26}$$

These quantify the causal *necessity* of vision (coarse) and interaction evidence (fine) for being correct on $x_i$.

**Proposition 1 ($D_c, D_f$ estimate causal necessity).** Under the standard assumption that the masking operation preserves non-interaction context and approximates the intended counterfactual intervention, $D_c(x_i)$ and $D_f(x_i)$ are Monte-Carlo estimators of $\text{ITE}_c(x_i)$ and $\text{ITE}_f(x_i)$, respectively.

*Proof sketch.* Each rollout contributes an unbiased sample of the correctness difference under the two conditions; averaging over $M$ rollouts estimates the expectation. □

Therefore, CSM is not "generic quality filtering": it explicitly retains samples with large causal effect of interaction evidence, aligning the training set with interaction-grounded reasoning.

### A.4. Why IRL Works: A Classical KL-Regularization Perspective Consistent with Interaction Reward

The Ego3S introduces an interaction-centric reward $r_i^{\text{int}}$ to reduce the reward–reality mismatch and force reliance on interaction cues. A standard and theoretically tractable instantiation is to include a counterfactual sensitivity term:

$$r_i^{\text{int}} = r_i^{\text{ans}} + \alpha\, s_i, \qquad s_i := \log p_\theta(o \mid x_i) - \log p_\theta(o \mid \tilde{x}_i), \tag{27}$$

where $o$ is the sampled output/trajectory, $x_i = (t_i, v_i)$, and $\tilde{x}_i = (t_i, \hat{v}_i)$ is the interaction-masked input.

**Lemma 3 (Classic exact equivalence to policy KL).** Let $\pi_\theta(o \mid x)$ be the policy distribution of outputs given input $x$. Then

$$\mathbb{E}_{o \sim \pi_\theta(\cdot \mid x_i)}\big[s_i\big] = \text{KL}\big(\pi_\theta(\cdot \mid x_i) \,\|\, \pi_\theta(\cdot \mid \tilde{x}_i)\big). \tag{28}$$

*Proof.* Expand the definition of KL divergence: $\text{KL}(\pi(\cdot \mid x)\|\pi(\cdot \mid \tilde{x})) = \mathbb{E}_{o \sim \pi(\cdot \mid x)}[\log \pi(o \mid x) - \log \pi(o \mid \tilde{x})]$. □

**Interpretation.** Maximizing the expected shaping term encourages the model to produce different output distributions when interaction evidence is removed, discouraging context-only shortcuts. When combined with CSM (which concentrates training on samples where interaction is necessary), this KL pressure becomes a targeted mechanism to align optimization with interaction grounding.

| Method | Time(h) ↓ | Egoplan2 | QAEgo4D | EgoTextVQA | EgoSchema | Egothink |
|---|---|---|---|---|---|---|
| Qwen2.5-VL-3b-Full | 46.5 | 18.43 | 53.76 | 11.28 | 42.57 | 49.58 |
| Qwen2.5-VL-3b-s1 (Muennighoff et al., 2025) (2025) | 38.4 | 17.89 | 54.42 | 10.83 | 42.98 | 49.31 |
| Qwen2.5-VL-3b-LIMO (Ye et al., 2025) (2025) | 39.7 | 18.54 | 54.17 | 11.19 | 42.46 | 49.23 |
| Qwen2.5-VL-3b-LIMR (Li et al., 2025d) (2025) | 60.9 | 19.38 | 55.23 | 11.64 | 43.27 | 49.86 |
| Qwen2.5-VL-3b-CSM (Ours) | 23.7 | 21.72 | 55.84 | 12.38 | 43.71 | 50.19 |
| Qwen2.5-VL-3b-GRPO (Guo et al., 2025) (2025) | 52.3 | 19.76 | 54.81 | 11.92 | 43.14 | 50.12 |
| Qwen2.5-VL-3b-RLOO (Muennighoff et al., 2025) (2025) | 56.8 | 18.12 | 53.24 | 10.37 | 41.86 | 48.47 |
| Qwen2.5-VL-3b-DAPO (Ye et al., 2025) (2025) | 48.4 | 18.81 | 54.63 | 11.57 | 42.49 | 50.22 |
| Qwen2.5-VL-3b-IRL (Ours) | 48.1 | 23.45 | 56.38 | 12.82 | 43.88 | 51.24 |
| **Qwen2.5-VL-3b-Ego3S (Ours)** | **25.6** | **25.50** | **60.00** | **13.80** | **45.2** | **55.43** |

*Table 3.* Performance on Qwen2.5-VL-3B. Upper: data selection methods. Lower: RL strategies. "Time" denotes training hours.

*Table 4.* Data Efficiency Analysis. Comparing the number of selected samples and their ratio relative to the full dataset (12k). The base model is the Qwen2.5-VL-7b.

| Method | Sample Size | Retention Ratio |
|---|---|---|
| Full Dataset | 12,000 | 100.0% |
| LIMO (Ye et al., 2025) | 3,604 | 30.0% |
| s1 (Muennighoff et al., 2025) | 3,765 | 31.4% |
| LIMR (Li et al., 2025d) | 5,015 | 41.8% |
| **Ego3S (Ours)** | **3,180** | **26.5%** |

### A.5. Why ECR Works: Variance as a Classical Proxy for Learning Signal in Policy Gradient

The ECR monitors reward variance $v_k(x)$ during training and refreshes data when it becomes uninformative. This aligns with standard variance-based notions of learning progress in stochastic gradient methods.

Let the per-sample policy-gradient (score-function) estimator be

$$g(x) := \mathbb{E}_{o \sim \pi_\theta(\cdot|x)} \big[ \nabla_\theta \log \pi_\theta(o \mid x)\, \hat{A}(o, x) \big], \tag{29}$$

where $\hat{A}(o, x)$ is an advantage estimate derived from rewards based on $r_i^{\text{int}}$.

**Lemma 4 (Variance controls gradient energy; classical bound).** Assume $\|\nabla_\theta \log \pi_\theta(o \mid x)\| \le G$ for all $o, x$. Then

$$\mathbb{E}_o \big[ \|\nabla_\theta \log \pi_\theta(o \mid x)\, \hat{A}(o, x)\|^2 \big] \le G^2\, \mathbb{E}_o \big[ \hat{A}(o, x)^2 \big] = G^2 \big( \mathrm{Var}_o[\hat{A}(o, x)] + (\mathbb{E}_o[\hat{A}(o, x)])^2 \big). \tag{30}$$

Hence, when the rollout variance $\mathrm{Var}_o[\hat{A}(o, x)]$ collapses across the current curriculum, the stochastic gradient energy diminishes, indicating reduced learning signal and potential stagnation.

**Connection to $v_k(x)$.** The $v_k(x)$ (reward variance across rollouts at training stage $k$) serves as a practical proxy for $\mathrm{Var}_o[\hat{A}(o, x)]$. Refreshing the curriculum when $v_k(x)$ becomes small reintroduces samples with larger counterfactual sensitivity (large $D_c, D_f$), thereby restoring informative gradients.

## B. Additional Comparative Results

### B.1. Generalization to Qwen2.5-VL-3B

To validate the scalability of Ego3S across model sizes, we conduct comprehensive experiments on the smaller Qwen2.5-VL-3B model.

### B.2. Comparisons with Data Selection Methods

We compare Ego3S against recent data selection methods on the 3B model. The results in upper part of Table 3 highlight two key findings: 1) Ego3S consistently outperforms the baseline of training on the full dataset across all benchmarks.

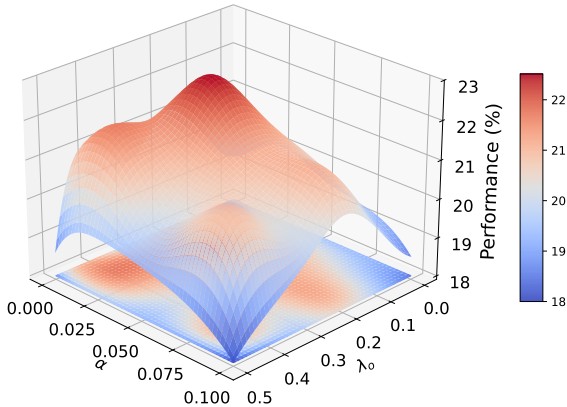

*Figure 11.* IRL performance varies with $\lambda_o$ and $\alpha$ on EgoTextVQA.

Remarkably, this superior performance is achieved while using only 26.5% of the data, reducing training time by 45%. This confirms that our findings on data efficiency generalize to smaller model scales. 2) Ego3S achieves a 3.12 points improvement over the latest method LIMR (Li et al., 2025d) on Egoplan2, while requiring only 42% of the training time. These results demonstrate that CSM's principle of prioritizing interaction-based reasoning samples remains effective even with reduced model capacity, as it isolates samples that genuinely require egocentric understanding rather than relying on model-specific characteristics.

### B.3. Analysis of Data Efficiency

We compare the data efficiency of Ego3S against recent state-of-the-art data selection methods on the Qwen2.5-VL-7b model. The statistics in Table 4 highlight two key findings: 1) Ego3S achieves robust reasoning capabilities using only **26.5%** of the full training data (3,180 out of 12,000 samples). This aggressive filtering significantly reduces computational overhead, cutting training costs by over **46%** compared to the full dataset baseline. This confirms that a vast majority of egocentric video data contains redundant background signals that contribute little to learning interaction logic. 2) Compared to the latest selection methods such as LIMO (Ye et al., 2025) (30.0%) and s1 (Muennighoff et al., 2025) (31.4%), Ego3S operates with the strictest selection budget yet yields superior results. Notably, our method achieves state-of-the-art performance while requiring only **63%** of the data volume used by the previous best method, LIMR (Li et al., 2025d) (3,180 vs. 5,015 samples).

These results demonstrate the effectiveness of our Counterfactual-based Selection Mechanism (CSM). Unlike methods that rely on general difficulty scoring (e.g., LIMO) or loss metrics (e.g., s1), CSM explicitly isolates samples that require *interaction-grounded reasoning*. By filtering out samples solvable via spurious context shortcuts, Ego3S proves that "less is more" when the remaining data is of high causal value.

### B.4. Comparisons with Recent RL Strategies

To validate the efficacy of our proposed Interaction-centric RL on smaller models, we compare Ego3S against several leading RL strategies. As shown in lower part of Table 3, we list the following conclusions: 1) Even without the proposed CSM, the IRL consistently outperforms the standard GRPO (Guo et al., 2025) across all benchmarks. For instance, on Egoplan2, IRL achieves 23.45% compared to GRPO's 19.76%, representing a 3.69% improvement. This improvement indicates that explicitly rewarding interaction grounding remains essential for guiding smaller models toward stable egocentric reasoning. 2) The proposed IRL also surpasses the state-of-the-art RL method DAPO (Yu et al., 2025) by 1.02 points on Egothink, which struggles to balance pretrained language priors and interaction-specific semantics even in smaller model scales.

## B.5. Comparison of Performance Improvements between First- and Third-person Perspectives

To further demonstrate the exclusive advantage of our Ego3S method, we conduct additional experiments on the third-person QA dataset NExT-QA (Xiao et al., 2021) using the same backbone and training schedule. As shown in Table 5, our Ego3S method yields a clear accuracy gain on QAEgo4D (Bärmann & Waibel, 2022). On the third-person NExT-QA (Xiao et al., 2021) benchmark, the improvement is marginal. This asymmetry is expected because Ego3S targets a failure mode that is especially severe in egocentric videos: inertial thinking, where models over-rely on linguistic and scene-level priors under dynamic, interaction-driven observations.

Concretely, the CSM filters out shortcut-solvable samples through counterfactual comparisons between multimodal and text-only predictions and between full and interaction-masked inputs, retaining instances where correct inference causally depends on localized interaction evidence. The IRL then mitigates the reward–reality mismatch by explicitly rewarding policies that ground decisions in these interaction cues, rather than in generic context priors. Finally, the ECR refreshes the curriculum based on reward variance to avoid training stagnation as the model's competence evolves. In third-person settings, the camera is exocentric and more stable, and global context is often coherent and predictive, so pretraining priors tend to align with the test distribution and shortcut reasoning is less harmful. As a result, hallucinations from inertial thinking are typically less prevalent, leaving less headroom for Ego3S to improve by suppressing shortcut reliance. In contrast, the larger gains on egocentric benchmarks suggest that Ego3S is most effective when correct decisions hinge on localized interaction evidence and global context is less predictive.

*Table 5.* Comparison of performance improvements between first- and third-person perspectives. QAEgo4D (Bärmann & Waibel, 2022) is an egocentric dataset, whereas NExT-QA (Xiao et al., 2021) is a third-person dataset. The column labeled "Rel. $\Delta$" indicates the relative improvement of Ego3S (Ours) over the baseline.

| Method | QAEgo4D (egocentric) | | NExT-QA (third-person) | |
|---|---|---|---|---|
| | Acc. (%) | Rel. $\Delta$ (%) | Acc. (%) | Rel. $\Delta$ (%) |
| Qwen2.5-VL-7b-Full | 63.80 | – | 71.1 | – |
| Qwen2.5-VL-7b-Ego3S (ours) | 66.80 | **+3.0** | 71.9 | **+0.8** |

# C. Hyperparameter Analysis

This section systematically analyzes the key hyperparameters of the Ego3S framework and validates their impact on model performance through ablation experiments on the EgoTextVQA dataset.

### C.1. IRL Parameters: $\lambda_o$ and $\alpha$

Figure 11 analyzes two core IRL hyperparameters on EgoTextVQA: $\lambda_o$ controls the minimum IoU requirement for interaction region attention, while $\alpha$ (penalty weight) balances the penalty strength for redundant object extraction.

### C.2. Impact of overlap threshold $\lambda_o$

The performance surface peaks at $\lambda_o \in [0.1, 0.3]$ and $\alpha \in [0.01, 0.05]$, reaching approximately 22-23% accuracy. When $\lambda_o$ is too low ($<0.1$), superficial attention allocation is permitted, degrading performance to approximately 18-19%. This indicates that overly lenient thresholds allow the model to satisfy constraints through surface-level attention without truly focusing on critical interaction regions. When $\lambda_o$ is too high ($>0.3$), despite reasonable interaction reasoning, excessively strict constraints limit attention flexibility, reducing performance to approximately 19-20%.

### C.3. Impact of penalty weight $\alpha$

Insufficient $\alpha$ ($<0.01$) permits indiscriminate object extraction, where the model tends to extract excessive irrelevant objects, introducing noise that interferes with decision-making. Excessive $\alpha$ ($>0.05$) overly constrains exploratory behavior, limiting the model's ability to discover potentially relevant objects, similarly impairing performance.

### C.4. Robustness analysis

A key advantage of Ego3S is its robustness to hyperparameter variations. The relatively broad optimal plateau—spanning $\lambda_o \in [0.1, 0.3]$ and $\alpha \in [0.01, 0.05]$—demonstrates that the method achieves consistently strong performance ($>21\%$

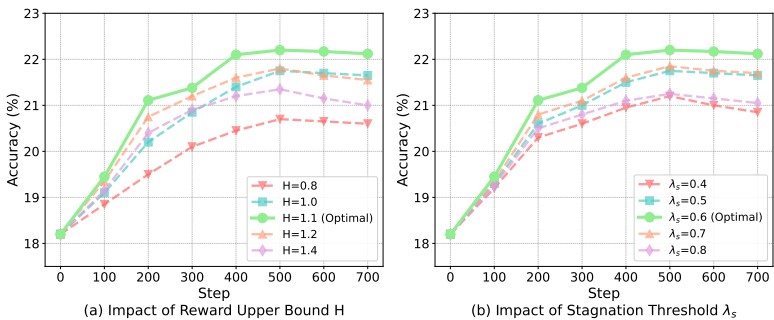

*Figure 12.* ECR hyperparameter ablations: (a) Reward upper bound $H$ impacts training dynamics and performance ceiling. (b) Stagnation threshold $\lambda_s$ affects curriculum refresh timing.

accuracy) across a wide range of reasonable settings. The robustness stems from grounding our reward mechanism in physically meaningful metrics (IoU overlap) rather than dataset-specific heuristics, enabling effective deployment across diverse egocentric scenarios with minimal tuning overhead.

### C.5. ECR Parameters: $H$ and $\lambda_s$

**Role of reward upper bound $H$.** Figure 12(a) examines the reward upper bound $H$ in ECR's recuration criterion. Setting $H = 1.0$ (perfect score) causes premature stagnation, as variance stabilizes before meaningful learning occurs. The model mistakenly believes it has reached optimality and stops exploring more challenging samples. $H = 1.1$ provides optimal headroom, allowing the model to stabilize near-ceiling performance while maintaining gradual improvement potential. This setting balances timely curriculum refresh with training stability. Higher values ($H \geq 1.2$) excessively delay recuration, missing opportunities to refresh the curriculum when current samples become uninformative, causing the model to waste computational resources on already-mastered data.

**Impact of stagnation threshold $\lambda_s$.** Figure 12(b) analyzes the stagnation threshold $\lambda_s$ for detecting learning plateaus. Too low $\lambda_s$ ($<0.5$) triggers excessive recuration, disrupting stable learning trajectories. Frequent dataset switching prevents the model from adequately learning the current curriculum, leading to suboptimal convergence. The optimal range $\lambda_s \in [0.5, 0.7]$ balances timely curriculum refresh with training stability. At this point, refresh is triggered only when approximately 60-70% of probing samples reach stable states, ensuring sufficient learning while avoiding excessive stagnation. Higher thresholds ($\lambda_s \geq 0.8$) delay recuration, causing extended periods of learning stagnation on exhausted samples, reducing learning efficiency.

## D. Qualitative Analysis

### D.1. CSM Selection Examples

To illustrate how CSM identifies samples requiring genuine interaction-centric reasoning, we present three representative cases in Figure 13.

In the bookshelf example, both *Text-only* and *Original* responses produce identical answers about storing and organizing books. The model succeeds purely through commonsense knowledge, as the shelf's function can be inferred from linguistic priors alone without observing the actual interaction.

Similarly, the ladder safety question yields matching answers about reduced overreaching and falling risks in both conditions. The model exploits pretraining knowledge rather than grounding its reasoning in the first-person perspective of standing on the ladder.

The envelope location example reveals more subtle bias. While both responses arrive at the correct answer "on the table," the *Text-only* response relies on generic patterns ("typically placed on a table"), whereas the *Original* response observes the actual office scene. This demonstrates that CSM filters samples where language shortcuts can accidentally produce correct answers, ensuring the model learns authentic visual-interaction understanding rather than exploiting linguistic correlations.

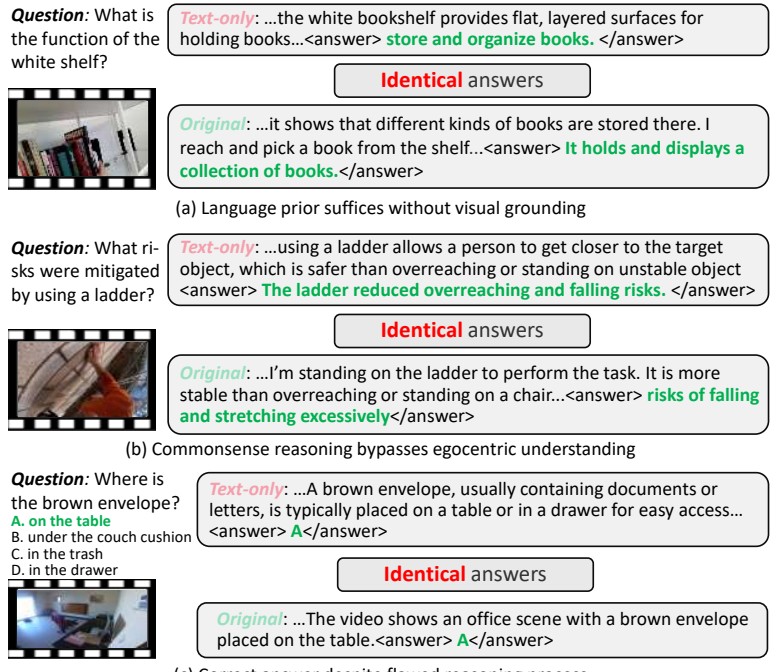

*Figure 13.* Three representative cases showing how CSM identifies samples requiring genuine interaction-centric reasoning.

## D.2. Examples of Global Context Bias

To illustrate how our CSM identifies and filters samples biased by global context, Figure 14 presents three cases where models produce correct answers without relying on interaction cues.

In the first example, when asked "Where was the Wi-Fi router?", both masked and original inputs yield the identical answer "at the table". With the key object (Wi-Fi router) masked and its exact location unobservable, the outcome is still inferred based on the surrounding context and the state of the table, demonstrating reliance on scene layout rather than interaction evidence.

The second example shows a similar pattern for "did I put the packed watermelon in the fridge?". The model responds correctly "yes" in both conditions. With the person's hands masked, the interaction action cannot be directly observed, yet the outcome can still be inferred solely from the refrigerator and its surrounding context.

The third example illustrates this in action reasoning. For "Why did I carry the monitor?", the model generates "Repair the monitor" regardless of masking. Even with the repair tool masked out, the outcome can be inferred solely from the object state and the workbench context, bypassing the need for grounding in the actual carrying action.

These cases reveal that models exploit static scene correlations to achieve correct answers without engaging interaction-specific evidence. Our CSM addresses this by computing fine-grained output discrepancy $D_f(x_i)$ to filter such samples, ensuring the retained data genuinely requires interaction-centric reasoning.

## D.3. Qualitative Analysis of Reasoning Quality

To better illustrate the advantages of our method, we present qualitative comparisons with both LIMR and GRPO across diverse egocentric reasoning scenarios.

**Comparison with LIMR.** As shown in Figure 15(a), LIMR fails to precisely identify the specific interaction context, instead providing a generic description (*"workbench"*) that overlooks critical spatial details like the *"green mat"* where operations occur. Similarly, in Figure 15(b), LIMR generalizes the action to *"washing dishes"* without recognizing the specific target object (*"pot"*) being manipulated. Figure 15(c) further demonstrates this pattern, where LIMR broadly describes *"items on shelf"* rather than accurately identifying the specific object being adjusted (*"pack of Ice Mountain Sparkling water"*). These

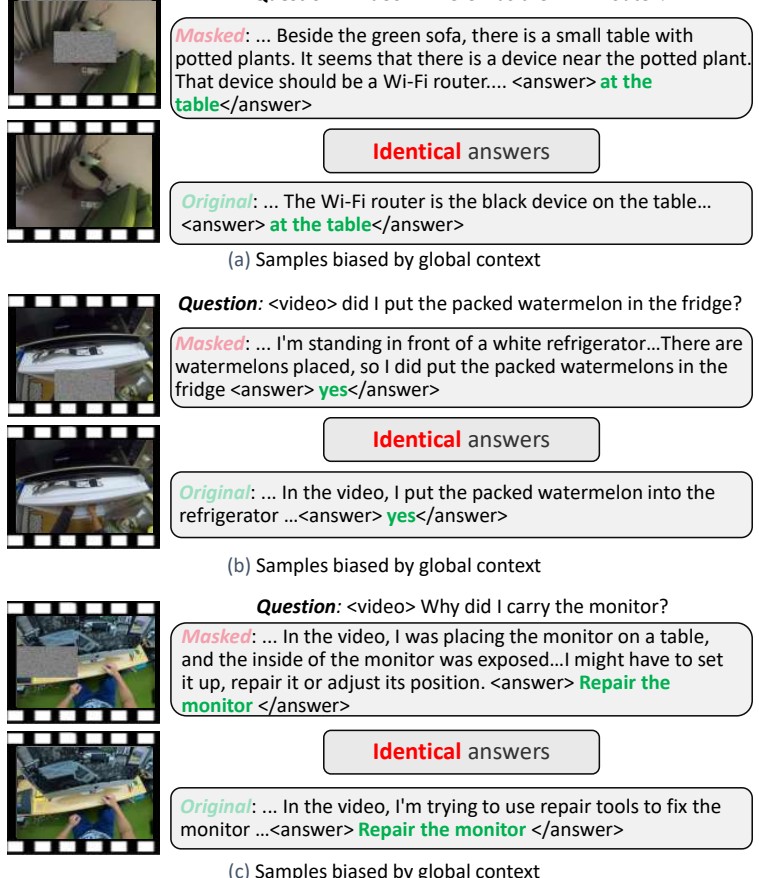

*Figure 14.* Illustrative examples of samples biased by global context.

examples reveal that LIMR suffers from *global context bias*—selecting samples that encourage models to rely on high-level scene understanding rather than fine-grained interaction-centric reasoning. In contrast, Ego3S consistently captures precise object attributes and interaction contexts through its CSM-selected high-value samples, demonstrating superior visual grounding capability.

**Comparison with GRPO.** Figure 16(a) illustrates a critical failure mode of GRPO: misidentifying the interaction object (*"lime or lemon"* instead of *"avocado"*), which cascades through the reasoning chain to produce an entirely incorrect conclusion. In Figure 16(b), GRPO fails to recognize *"ice cubes"* and consequently misinterprets the action as *"retrieving an item"* rather than the actual behavior of *"putting ice into a bottle"*. Figure 16(c) demonstrates another failure: incorrectly judging two identical tools held in opposite hands as *"different tools"* due to insufficient attention to visual interaction cues. These cases expose GRPO's vulnerability to *reward hacking*—achieving correct answers in some cases without genuine visual grounding, as the binary correctness signal fails to enforce interaction-aware reasoning. Our Ego3S, equipped with interaction-centric rewards ($r^{int}$ and $r^p$), consistently produces reasoning chains that explicitly reference observable interaction evidence, ensuring both accuracy and interpretability.

*Question*: Where was the activity of disassembling and soldering conducted?

*Groundtruth:* At a cluttered wooden desk with a green mat.

*LIMR*: <think>... The table is equipped with knives such as cutting pads, cotton swabs, brushes and items related to Polaroid cameras. **It seems to be a workbench specifically designed for manual operation.** </think> <answer> **On a workbench.** </answer>

*Ego3S (Ours)*: <think>… And all kinds of tools were piled up on the table, **my main operations were all done on a green mat** …</think> <answer> **On a green mat on the table** </answer>

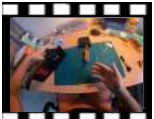

(a)

*Question*: What am I doing?
*Groundtruth:* washing a pot.

*LIMR*: <think> ... I'm in a kitchen, and I see a sink with various items around it. There's a pot, a dish, and some cleaning supplies. I'm holding a yellow sponge and a glass. Hmm, **I seem to be in the process of washing dishes.** </think> <answer> **washing dishes** </answer>

*Ego3S (Ours)*: <think> ... My left hand was holding the handle of the black pot, which was filled with water, **and my right hand was working the yellow sponge over it as I continued washing the pot.**</think><answer> **washing a pot** </answer>

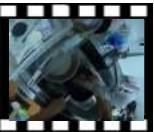

(b)

*Question*: What were you adjusting in the dimly lit room ?
*Groundtruth:* a pack of water bottles

*LIMR*: <think>… There are also other items like a package of Pampers Aqua Pure and a container of water nearby. **It seems that I'm sorting out all kinds of items on the shelves**</think> <answer> **The items on shelf.** </answer>

*Ego3S (Ours)*: <think> .... **The video shows my hands adjusting a pack of Ice Mountain Sparkling water on the shelf**. Let me check again that I didn't adjust anything else on the shelf </think> <answer> **a pack of Ice Mountain Sparkling water on a shelf.** </answer>

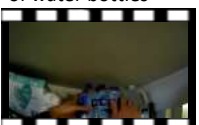

(c)

*Figure 15.* Qualitative comparison between Ego3S and LIMR.

**Question**: What am I doing?
**Groundtruth:** peeling the avocado

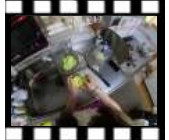

**GRPO**: <think>…The video shows me in the kitchen, I am handling a green fruit ,likely a lime or lemon, **probably to make a drink like lemonade or a cocktail.**</think> <answer> **I am making a drink** </answer>

**Ego3S (Ours)**: <think> …. There is also a fruit knife placed beside the sink, and there are vegetables on the sink that look like they have been processed…**The avocado in my hand has been cut open and I'm peeling it.** </think> <answer> **I'm dealing with avocados.** </answer>

(a)

**Question**: What am I doing?
**Groundtruth:** Putting ice into a bottle.

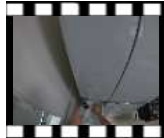

**GRPO**: <think>I am reaching into a cabinet**, possibly to retrieve or place an item.** The presence of a cylindrical object in my hand suggests I might be handling a container or a bottle…</think><answer> **I am retrieving an item from the cabinet.**</answer>

**Ego3S (Ours)**: <think> … There are some transparent solids stored in the cabinet, **which seem to be ice cubes**… My other hand was holding a white cylindrical bottle…**I'm putting ice cubes into it.** </think><answer> **removing ice cubes from the cabinet and putting them back into the bottle** </answer>

(b)

**Question**: Are the item I'm holding in my left hand the same type as the item I'm holding in my right hand?

**Groundtruth:** yes

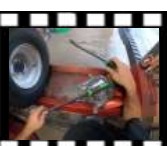

**GRPO**: <think>… The item in my left hand appears to be a long, thin object with a pointed end, which looks like a file. **The item in my right hand is also long and thin, but it has a different shape and appears to be a different tool.** </think> <answer>**No.** </answer>

**Ego3S (Ours)**: <think>… the same shape and colors of black and green, matching each other. **The two seem to be tools of the same type, like pries or screwdrivers**…</think> <answer> **Yes.** </answer>

(c)

*Figure 16.* Qualitative comparison between Ego3S and GRPO.

