# OpenReview forum: "Ego3S: Select, Strengthen, and Synchronize for Efficient Egocentric Reasoning"
_ICML.cc/2026/Conference — ICML 2026 regular_

### Official Review · Reviewer_oZZ4 · 2026-03-05

**Soundness:** 3
**Presentation:** 4
**Significance:** 3
**Originality:** 3
**Overall Recommendation:** 4
**Confidence:** 4

**Summary:**

The paper introduces Ego3S, a framework designed to improve the efficiency and accuracy of Large Vision-Language Models (LVLMs) in egocentric reasoning tasks. The authors identify a key challenge termed "inertial thinking," where models rely too heavily on language priors and global context rather than immediate, localized visual cues. To combat this, Ego3S employs a three-stage pipeline: (1) Causal-based Sample Mining (CSM), which uses counterfactual analysis to identify high-value samples where visual interaction is critical; (2) Interaction-centered Reinforcement Learning (IRL), which rewards the model for grounding its reasoning in specific interaction regions; and (3) Ecological Curriculum Regularization (ECR), which dynamically adjusts the training difficulty based on reward variance.

**Compliance With Llm Reviewing Policy:**

Affirmed.

**Final Justification:**

Most of my concerns have been satisfactorily addressed. I' ll maintain my positive score.

**Key Questions For Authors:**

How does the performance of the IRL module degrade if the detected "Interaction Regions" are inaccurate or partially occluded? Since egocentric video is notoriously shaky and prone to motion blur, understanding the robustness of the reward signal is crucial.

Does the CSM-filtered training lead to any performance drop in general VQA tasks that do not involve direct egocentric interaction? I would like to see if the model still maintains strong "global" understanding after being heavily "strengthened" on localized cues.

In the IRL stage, how do you ensure the model doesn't simply learn to mention the "Interaction Region" keywords to get a higher reward without actually performing the causal reasoning? Did you observe any instances of "correct answer, wrong reason" during the RL process?

Have you tested whether the benefits of Ego3S (the 26% data efficiency) hold as you scale the base model to much larger parameters (e.g., 34B or 70B)? Knowing if this efficiency gain is model-size-agnostic would significantly increase the paper's impact.

**Limitations:**

See Cons.

**Strengths And Weaknesses:**

**Strengths**

1.The technical design is well-motivated by the unique challenges of egocentric vision. The use of counterfactual intervention to identify "visually-dependent" samples is a principled way to reduce data redundancy and address the bias of language priors.

2. The paper is exceptionally well-written. The distinction between third-person and first-person reasoning is clearly articulated, and the figures effectively illustrate the proposed pipeline. The inclusion of theoretical derivations in the appendix adds significant rigor.

3. As egocentric vision is a cornerstone of Embodied AI, the ability to train models that are both more accurate and more efficient is highly valuable. The finding that a fraction of the data can yield superior results is an important contribution to the community focusing on efficient LLM/VLM training.

4. While the individual components (RL, Curriculum Learning) are known, their integration into a unified "Select, Strengthen, and Synchronize" framework specifically tailored to mitigate "inertial thinking" in egocentric contexts is novel and creative.

**Weaknesses**

1. Dependency on Interaction Priors: The IRL stage relies on the identification of "Interaction Regions." While the paper describes how these are obtained, the performance of the model may be sensitive to the quality of these spatial priors. A more thorough discussion on how the framework handles noisy or absent interaction cues in highly cluttered environments would be beneficial.

2. Potential for Overfitting to Training Distribution: The CSM module filters out a large portion of "easy" or "context-heavy" samples. While this improves efficiency, there is a minor concern that the model might lose some general reasoning capabilities or common-sense priors that are actually helpful in broader, non-interaction-heavy scenarios.

3. Inference Efficiency vs. Training Efficiency: While the paper demonstrates impressive training efficiency (46% reduction in cost), it remains unclear if the resulting model offers any inference-time advantages (e.g., shorter reasoning chains) compared to models trained on the full dataset.

---

> ### Author Rebuttal · Authors · 2026-03-31
>
> **Response for R# oZZ4**
>
> *Rn-Wn(Qn): Response n for Weakness n (Key Question n)*
>
> Thanks for your insightful comments.
>
> **R1-(W1&Q1):**
> Please refer to our response to **Reviewer TNRo  (R3-Q1). The robustness experiment in Tab.3 shows that Ego3S consistently outperforms GRPO when interaction regions are partially masked.**
>
> In cluttered scenes with noise, IRL is an **auxiliary shaping reward built on top of the original correctness reward, rather than a replacement for it**. Therefore, inaccurate interaction regions may weaken the effect of $r_{int}$, instead of fundamentally corrupting the RL objective.
>
> When IRL becomes less informative under such noise, the other two components of Ego3S can still contribute in distinct ways. CSM continues to filter out samples that can be solved by language priors or global context. ECR can still monitor whether the current curriculum remains informative and refresh it when it becomes less useful. **Thus, even if the interaction reward is weakened, Ego3S still achieves interaction-guided reasoning and curriculum adaptation, thus focusing on key interaction cues and stable optimization**.
>
> ------
> **R2-(W2&Q2):**
> To address the concern about overfitting, please refer to our response and added experiment in **Reviewer TNRo R1-(W1&W2&Q3) and Tab.1**, which shows that **CSM does not suppress correct predictions that lie within model’s prior distribution**.
>
> In addition, to examine whether CSM-filtered training harms performance on general VQA tasks, we report results in **Appendix B.5, Table 5, where Ego3S outperforms the baseline on third-person VQA**.
>
> **These results show that Ego3S improves interaction-based reasoning while preserving general reasoning ability**. A likely reason is that CSM does not distort the training distribution. Instead, it selectively removes samples that can already be solved primarily by language priors or global context alone. Such samples add little supervision, as their knowledge is likely already learned during pretraining. **In contrast, samples seleted by CSM still require not only localized interaction cues but also sufficient global understanding to resolve the task. Thus, the filtered set still exercises general visual reasoning while more effectively grounding the model’s predictions in actual interaction evidence.**
>
> In summary, Ego3S improves egocentric reasoning without compromising broader visual-language understanding.
>
> ------
> **R3-W3:**
> *We agree that inference-time efficiency is another important direction*, **but it iis orthogonal to our focus and would require adaptive decoding or token compression. Reducing training data does not shorten inference time, as inference cost is mainly determined by the model architecture and decoding strategy**.
>
> We aim to improve egocentric reasoning more efficiently during training. Ego3S achieves better performance using only 26.5% of training data while reducing training cost by over 46%. *At inference time, CSM is not involved in decoding and thus adds no overhead (average inference time per sample: CSM: 3.25s vs Full: 3.31s).*
>
> Thank you for pointing out this, which is a promising direction for future work.
>
> ------
> **R4-Q3:**
> In fact, this issue is explicitly considered in our RL design. **To prevent model from exploiting interaction-centric reward by overly mentioning interaction-related cues, we introduce the penalty term $r_p$**. While the $r_{int}$ guides the model to reason based on objects in the interaction region, the penalty term $r_p$ penalizes redundant object enumeration. **If the policy attempts to cheat by generating an unnecessarily long list of objects just to hit the target keyword, $r_p$ imposes a heavy negative reward.** Hence, the optimal policy must learn to reason based on correct interaction cues, rather than indiscriminately listing candidate objects.
>
> *To address your concern, we also analyze “correct answer, wrong reason” (CAWR) cases during training.* We use GPT5.4 as a judge to track CAWR cases across 3,000 training samples.
>
> **See Tab.4:** https://anonymous.4open.science/r/Ego3S-rebuttal-8536.
>
> **Tab. 4 shows that IRL does not increase CAWR during training and consistently outperforms GRPO.** Moreover, $r_p$ prevents reward hacking through redundant object enumeration, thus encouraging reasoning based on interaction cues.
>
> ------
> **R5-Q4:**
> Due to compute constraints (8×A6000), we have not yet completed RL training at 34B/70B scale. *To partially address this, we add an experiment on a larger model Qwen3-VL-8B.*
>
> **See Tab.5:** https://anonymous.4open.science/r/Ego3S-rebuttal-8536.
>
> The results show that Ego3S still maintains an advantage on this larger backbone while using only a part of training data. This suggests that its benefit is **not tied to a specific smaller model**, but stems from the quality of selected interaction-dependent samples. This provides initial evidence that Ego3S’s data-efficiency gains generalize to larger backbones.

---

> > ### Author Rebuttal · Reviewer_oZZ4 · 2026-04-04
> >
> > Most of my concerns have been satisfactorily addressed. I' ll maintain my score.

---

> > > ### Author Response · Authors · 2026-04-04
> > >
> > > Response 2 for R# oZZ4:
> > >
> > > Thank you very much for your recognition and for your valuable feedback. We are glad that our rebuttal has adequately addressed your main concerns. We also sincerely appreciate your constructive suggestions regarding the scope and presentation of our contribution. In the final version, we will carefully revise the paper according to your comments.

---

### Official Review · Reviewer_TNRo · 2026-03-09

**Soundness:** 3
**Presentation:** 2
**Significance:** 3
**Originality:** 2
**Overall Recommendation:** 5
**Confidence:** 2

**Summary:**

The paper proposes Ego3S, a training framework designed to improve egocentric reasoning in large vision–language models (LVLMs). Egocentric reasoning refers to understanding events from a first-person perspective, where key information often comes from localized human–object interactions rather than global scene context. The authors observe that existing LVLMs often suffer from “inertial thinking”, where models rely heavily on language priors or common contextual patterns instead of the actual visual evidence in the video. This leads to incorrect predictions when the correct answer depends on interaction cues in the current frame. To address this problem, the paper introduces a three-stage training framework that improves reasoning by focusing the model on interaction evidence and filtering out misleading training signals.

**Compliance With Llm Reviewing Policy:**

Affirmed.

**Key Questions For Authors:**

1. How robust is the framework to errors in interaction region detection? If the interaction region is incorrectly identified, how does it affect the RL reward and overall training stability?
2. Can the authors clarify how Ego3S differs conceptually from existing data curation or curriculum learning approaches, and whether similar gains could be achieved by simpler sample filtering strategies?
3. Could the authors provide a more detailed explanation of the interaction-centric reward design? In particular, how does the reward formulation prevent the model from exploiting prior distributions (e.g., typical VQA shortcuts) and ensure that the reasoning process truly relies on interaction evidence?

**Limitations:**

The framework mainly improves data selection, reward design, and curriculum scheduling, while the underlying vision-language model architecture remains unchanged. Therefore, the contribution is largely training-pipeline and data-centric, rather than introducing new model architectures or reasoning mechanisms. The method relies on detecting interaction regions (e.g., via similarity measures). If the interaction region is incorrectly identified, the reward signal and training guidance may become unreliable, affecting model performance.

**Strengths And Weaknesses:**

Strengths
1. The paper proposes an important problem: egocentric understanding requires action understanding due to dynamic viewpoints and localized interaction cues. Models often rely on language priors or global context instead of visual interaction evidence, making this a meaningful research problem.

2. The paper proposes the Ego3S framework with three complementary components:

    * Select: Counterfactual-based Selection Mechanism (CSM) filters samples that can be solved by language priors or global context.

    * Strengthen: Interaction-centric Reinforcement Learning (IRL) rewards models for The CLIP similarity for interactiongrounding reasoning in interaction cues.

    * Synchronize: Evolving Curriculum Recuration (ECR) dynamically updates the training dataset during learning.

   This pipeline systematically addresses both data quality and training signals for egocentric reasoning.

3. Efficient training with less data
The method achieves better performance using only 26.5% of the training data, while reducing training cost by over 46%, demonstrating strong data efficiency.

Weakness
1. The explanation of the interaction-centric reward can be improved. The current annotation is confusing: the index i is used both as the sample index in GRPO and as the VQA sample index in Section 3.1. In addition, the paper does not explicitly mention that the reward uses the largest object similarity in the answer, which makes the formulation difficult to understand.

    After reading the code, the idea becomes clearer: the reward appears to assign lower rewards to objects that are overly common answers for VLMs, presumably to encourage interaction-focused reasoning. However, it remains unclear how strongly this mechanism correlates with the intended interaction-centric objective.

    Although the paper includes experiments demonstrating improved utilization of interaction cues, the motivation and formulation of this reward would benefit from a clearer explanation

2. It is unclear whether the interaction-centric reward could harm performance for interactions that already lie within the LLM/VLM prior distribution. In such cases, penalizing frequent answers might suppress correct predictions. The paper should investigate how the method balances the bias–variance tradeoff introduced by this reward design.

3. The paper didn't verify if CLIP similarity reality selects the best key frames for interaction query, it would be better to provide such details.

---

> ### Author Rebuttal · Authors · 2026-03-31
>
> **Response for R# TNRo**
>
> *Rn-Wn(Qn): Response n for Weakness n (Key Question n)*
>
> Thanks for your insightful comments.
>
> **R1-(W1&W2&Q3):**
>
> *1. Notation Clarity*
>
> First, we apologize for notational ambiguity. We will revise the notation by reserving $i$ for the training-sample index and using $m$ for rollout index, so that $o_{i,m}$ denotes the $m$-th rollout for the $i$-th instance.
>
> *2.Clarifying the Largest Similarity*
>
> Second, we do not explicitly state that the reward is determined by the best-matching object. However, **your intuition perfectly aligns with our design**. For each rollout, we extract object mentions from the response and compute their similarity to the interaction region. The interaction reward is activated when *at least one* mentioned object exceeds the threshold. **Thus, our formulation is mathematically equivalent to using largest object–interaction similarity to verify interaction.**
>
> *3.Prior Shortcuts*
>
> Moreover, **IRL does not explicitly penalize common answers, but to encourage interaction-grounded reasoning**. In Ego3S, shortcut reliance from language priors or global context is mainly reduced by CSM, which removes samples solvable by shortcuts, thus focusing on interaction cues. IRL addresses this from a various view: under correctness-only RL, interaction-based reasoning and shortcut-based reasoning receive the same rewards. **IRL mitigates this by assigning an additional reward only when mentioned objects are aligned with interaction regions, guiding models to attend to interactions**.
>
> *4.Bias-Variance Tradeoff & Prior-Aligned Interactions*
>
> The above also explains why our reward should not be seen as a penalty for frequent answers or objects. The $r_{int}$ is an extra reward for grounded interaction mention, **not a subtraction from the original correctness reward. A correct answer does not lose its base reward simply because it is common**. Likewise, the penalty $r_p$ is **not to suppress prior knowledge**: it prevents model from obtaining $r_{int}$ by overly outputting irrelevant objects, rather than learning to use interaction cues for reasoning.
>
> To verify that IRL does not suppress correct predictions that lie within VLM prior distribution, we add an experiment on Egoplan2 subset whose groundtruth belong to top 10% most frequent objects in third-person dataset CoCo2017 & our first-person training dataset.
>
> **See Tab.1:** https://anonymous.4open.science/r/Ego3S-rebuttal-8536
>
> Tab.1 shows that Ego3S balances bias-variance tradeoff: it does not suppress common correct answers and still guide model toward interactions, avoiding degradation on common objects.
>
> ------
> **R2-W3:**
>
> As shown in Sec.3.1, we localize interaction region within a frame rather than an entire frame. We apply a sliding window to generate candidate regions, compute CLIP similarity between each crop & query, and select the crop with the highest similarity as interaction region.
>
> To verify the quality, we add an experiment with 100 manually annotated interaction boxes. We compare random crop, center crop and our method using accuracy & mean IoU.
>
> **See Tab.2:** https://anonymous.4open.science/r/Ego3S-rebuttal-8536
>
> Tab.2 suggests that our CLIP-based method is reliable. Moreover, **0.58 Mean IoU** shows close alignment with human-annotated boxes.
>
> ------
> **R3-Q1:**
>
> To address your concern, we add a robustness experiment by randomly masking 10% of interaction region in raw frame and evaluating under this noise.
>
> **See Tab.3:** https://anonymous.4open.science/r/Ego3S-rebuttal-8536
>
> *Tab.3 shows that Ego3S consistently outperforms GRPO. We also compare noisy learning curves in* **Fig.2** (https://anonymous.4open.science/r/Ego3S-rebuttal-8536). Despite minor fluctuations, the trend remains consistent, showing that interaction-centric reward does not destabilize training.
>
> Under moderate occlusion, the worst case is usually a weaker or missing interaction bonus, rather than a penalty on correct behavior. Thus, even if IRL weakens, the base reward is preserved. CSM/ECR still function normally, so Ego3S remains stable.
>
> ------
> **R3-Q2:**
>
> **Existing methods typically define data quality by difficulty,  but cannot distinguish whether a sample activates interaction-based reasoning. Ego3S departs from this paradigm by explicitly targeting interaction-based reasoning.** CSM retains only samples that truly require attention to interaction cues, removing samples that overly rely on priors.
>
> As shown in Table 1 of paper, Ego3S outperforms existing methods. **Figure 10(b) already provides a comparison with difficulty-based filtering**. Such show that *training on merely hard samples performs worse than interaction-dependent samples from CSM*.
>
> ------
> **R4-limitation:**
> We agree that Ego3S is not a new architecture. However, it is adaptable to various models: our cross-model transfer results show that Ego3S generalize across distinct LVLMs and our robustness analysis shows that it remains stable under noise.

---

> > ### Author Rebuttal · Reviewer_TNRo · 2026-04-01
> >
> > This rebuttal responds to the main concerns in a fairly substantial way, especially regarding the motivation of the interaction-centric reward, whether it harms frequent correct answers, whether CLIP-based region selection is reliable, and whether errors in interaction-region detection destabilize training. As a result, it significantly improves the credibility of the paper. However, the contribution remains primarily at the level of training pipeline and data curation rather than model architecture. I will increase the score accordingly.

---

> > > ### Author Response · Authors · 2026-04-01
> > >
> > > **Response 2 for R# TNRo**:
> > >
> > > Thank you very much for your recognition and for your thoughtful and valuable feedback. We are glad that our rebuttal has adequately addressed your main concerns and improved the credibility of the paper. We also sincerely appreciate your constructive suggestions regarding the scope and presentation of our contribution. In the final version, we will carefully revise the paper according to your comments to further clarify the methodological contribution and improve the overall presentation.

---

### Official Review · Reviewer_XrfR · 2026-03-09

**Soundness:** 3
**Presentation:** 3
**Significance:** 3
**Originality:** 3
**Overall Recommendation:** 4
**Confidence:** 3

**Summary:**

This paper studies the challenge of egocentric reasoning in large vision-language models, highlighting that egocentric scenes are dynamic and rely on localized interaction cues rather than global context. To address the tendency of existing models to over-reliance on language priors, the authors propose Ego3S, a three-stage framework that includes counterfactual-based data selection, interaction-centric reinforcement learning rewards, and a variance-aware training schedule. Experiments demonstrate the effectiveness of Ego3S, which also significantly reduces computational cost.

**Compliance With Llm Reviewing Policy:**

Affirmed.

**Final Justification:**

Thank you for the rebuttal—my questions have all been fully addressed.

**Key Questions For Authors:**

Please see the weakness.

**Limitations:**

yes

**Strengths And Weaknesses:**

## Strengths

1. The motivation of the paper is intuitive and well-grounded. The problem formulation and the proposed solution appear reasonable and interesting, and the overall framework logically addresses the limitations of existing approaches.

2. The paper includes extensive experiments with detailed discussions and comparisons against existing methods. These empirical results provide fairly comprehensive evidence supporting the effectiveness of the proposed approach.

3. In addition to empirical validation, the paper also provides a theoretical analysis that helps explain why the proposed method works, which further strengthens the technical soundness of the approach.

---

## Weaknesses

1. I have some concerns regarding the third component of the method (Sect. 3.3. Evolving Curriculum Recuration) that relies on **variance-based data selection**. If the selection process depends entirely on variance, it may lead to a newly constructed training dataset whose reward quality is potentially worse than that of the original dataset. Conceptually, it might be more reasonable to design this mechanism as a form of **bias–variance trade-off** rather than relying solely on variance. It would be helpful if the authors could discuss this issue further and possibly provide additional analysis or experiments to justify or improve this design.

2. The time evaluation reported in Table 1 appears somewhat unusual. For example, the training time using the full dataset is reported to be **lower than that of an existing method such as Qwen2.5-VL-7B-LIMR (2025)**, which seems counterintuitive. The authors should clarify the reason for this result. Additionally, the exact protocol used to measure training time should be explained in more detail.

3. The paper provides an anonymous code repository, but the link appears to have expired. This prevents verification of the implementation and experimental details. The authors should clarify the status of the repository and provide an accessible link if possible.

---

> ### Author Rebuttal · Authors · 2026-03-31
>
> **Response for R# XrfR**
>
> *Rn-Wn(Qn): Response n for Weakness n (Key Question n)*
>
> We sincerely appreciate your valuable comments.
>
> **R1-W1:**  Thanks for your valuable advice. We would like to clarify a potential misunderstanding regarding the role of ECR and provide a supplementary experiment according to your advice.
>
> *1.ECR is a trigger for refreshing the training curriculum, not a selector of data quality*
>
> The ECR does not use reward variance as a metric to define data quality or to directly select new informative training samples. Instead, as mentioned in Sec. 3.3, *ECR uses the dynamic change in reward variance solely **as a trigger signal** to determine **when the current curriculum has become uninformative and training has reached stagnation***. Once stagnation is detected, the new training set **is not** constructed by variance-based selection; rather, **it is rebuilt by re-applying the Counterfactual-based Selection Mechanism (CSM) to the full data pool using the updated model**. Therefore, the quality of the refreshed curriculum is guaranteed by CSM, which retains samples that genuinely require interaction-grounded reasoning rather than shortcut-based inference. The ECR relies on variance only as a signal of whether the current curriculum still provides sufficient learning signal, thereby continuously pushing the performance upper bound. This effect of ECR is also supported by the results in **Figure 9(a) and Table 6 (No.5 vs No.6)** of paper.
>
> *2.Additional experiment: bias–variance combined ECR trigger*
>
> To further address your concern, *we have conducted an additional experiment comparing our original ECR with a bias–variance combined variant ECR-b.* Specifically, we extend ECR to trigger curriculum refresh only when both the **mean reward change** ($\Delta_\text{Mean}$) and the **reward variance stagnation signal** indicate convergence:
> (1) the mean reward change is below the threshold ($\Delta_\text{Mean} < 0.1$); and
> (2) the variance-based stagnation score on the probing set $S_k(D_{\mathrm{probe}})$, exceeds the $\lambda_s$, following the original ECR.
>
> |Method|Trigger Frequency|Acc.%|
> |---|---:|---:|
> |w/o ECR|0|22.00|
> |ECR-b|3|22.30|
> |**ECR (Ours)**|3|**22.40**|
>
> As shown in the table, *the original ECR achieves slightly better performance than the bias–variance combined variant ECR-B using Qwen2.5VL-7b on EgoTextVQA*. Moreover, the trigger frequency also remains identical. Such results show that the original ECR is already sufficient to capture curriculum stagnation.
>
> A likely reason is that, after CSM filters out shortcut-solvable and noisy samples, the remaining training curriculum mainly consists of informative examples with relatively reliable reward signals. **In this setting, when the reward variance becomes consistent across different training stages, it typically indicates that the model’s responses to the current curriculum have also become stable, meaning that most of the available learning signal from these samples has already been extracted. In other words, the expected mean reward is also likely to remain nearly unchanged. Therefore, notable shifts in reward variance are usually accompanied by corresponding changes in the expected mean reward.** This explains why the original variance-only ECR is already sufficient to effectively detect curriculum stagnation.
>
> **R2-W2:**  We appreciate your valuable suggestion. Below, we clarify the exact protocol used to measure computational time and explain why LIMR incurs such a high cost.
>
> The time in Table 1 of paper denotes the **total computation cost**, i.e., **data selection + RL training**, rather than only the final training stage. As shown in Fig.1(https://anonymous.4open.science/r/Ego3S-rebuttal-8536), the full-data baseline costs 60.5 hours (training on all 12,000 samples). LIMR, although trained on only 5,015 selected samples, is more time-consuming overall. This is because its sample scoring depends on the model’s learning trajectories and therefore requires an additional full training process before subset construction. Specifically, it first performs a learning-impact-based selection stage, which includes full-data training and selection (**60.5 + 2.5 hours**), and then still requires **40.2 hours** of RL training on the selected subset, resulting in a total of **103.2 hours**. Therefore, **LIMR only reduces data redundancy**, whereas our method improves both **data efficiency** and **training efficiency**.
>
>
>
> **R3-W3:**  We sincerely apologize for the outdated link. The code repository has remained accessible. The problem is that the link in the manuscript was not updated correctly. The valid link is the one that has already been provided on the OpenReview abstract page  (which Reviewer TNRO noted that “after reading the code, the idea becomes clearer"):
>
> **Code**: https://anonymous.4open.science/r/Ego3S-70A2.
>
> We will correct the URL in the final version.
>
> We will add the results and discussions in the final version.

---

> > ### Author Rebuttal · Reviewer_XrfR · 2026-04-03
> >
> > Thank you for the rebuttal—my questions have all been fully addressed. I have increased my score to **4**.

---

> > > ### Author Response · Authors · 2026-04-04
> > >
> > > Response 2 for R# XrfR:
> > >
> > > Thank you very much for your recognition and for your valuable feedback. We are glad that our rebuttal has adequately addressed your main concerns and improved the credibility of the paper. We also sincerely appreciate your constructive suggestions regarding the scope and presentation of our contribution. In the final version, we will carefully revise the paper according to your comments.

---

### Decision · Program_Chairs · 2026-04-30

**Decision:**

Accept (regular)

**Comment:**

This paper was reviewed by three experts in the field, resulting in recommendations of (Weak Accept x 2, Accept). Based on this feedback, the decision is to recommend acceptance. The reviewers raised several valuable concerns that must be addressed in the final camera-ready version, particularly regarding more detailed experimental comparisons and ablation studies (raised by Reviewers XrfR and oZZ4), and clearer paper presentation and statements (raised by all three reviewers). The authors are encouraged to address these points to the best of their ability.